

# A comparison of carbon monoxide retrievals between the MOPITT satellite and Canadian High-Arctic ground-based NDACC and TCCON FTIR measurements

Ali Jalali[1], Kaley A. Walker[1], Kimberly Strong[1], Rebecca R. Buchholz[2], Merritt N. Deeter[2], Debra Wunch[1], Sébastien Roche[1], Tyler Wizenberg[1], Erik Lutsch[1], Erin McGee[1], Helen M. Worden[2], Pierre Fogal[1], and James R. Drummond[3]

[1]Department of Physics, University of Toronto, Toronto, ON, Canada

[2]Atmospheric Chemistry Observations and Modeling Laboratory, National Center for Atmospheric Research, Boulder, CO, U.S.A.

[3]Department of Physics and Atmospheric Physics, Dalhousie University, Halifax, NS, Canada

*Correspondence to:* Kaley A. Walker (kaley.walker@utoronto.ca)

**Abstract.** Measurements of Pollution in the Troposphere (MOPITT) is an instrument on NASA's Terra satellite that has measured tropospheric carbon monoxide (CO) from early 2000 to the present day. Validation of data from satellite instruments like MOPITT is often conducted using ground-based measurements to ensure the continued accuracy of the space-based instrument's measurements and its scientific results. Previous MOPITT validation studies generally found a larger bias in the

5   MOPITT data poleward of 60 °N. In this study, we use data from 2006 to 2019 from the Bruker IFS 125HR Fourier Transform Infrared spectrometer (FTIR) located at the Polar Environment Atmospheric Research Laboratory (PEARL) in Eureka, Nunavut, Canada to validate the MOPITT Version 8 retrievals. These comparisons utilize mid- and near-infrared FTIR measurements made as part of the Network for the Detection for Atmospheric Composition Change (NDACC) and the Total Carbon Column Observing Network (TCCON), respectively. All MOPITT version 8 retrievals within a 1° radius from the

10   PEARL Ridge Laboratory and within a 24-hour time interval are used in this validation study. MOPITT retrieval products include those from the near-infrared (NIR) channel, the thermal infrared (TIR) channel, and a joint product from the thermal and near-infrared (TIR-NIR) channels. Each channel's detector has four pixels. We calculated the MOPITT pixel-to-pixel biases for each pixel, which were found to vary based on the season and surface type (land or water). The systematic bias for pixel 1 over land is larger than that for other pixels, which can reach up to 20 ppb. We use a small-region approximation method to





find filtering criteria. We then apply the filters to the MOPITT dataset to minimize the MOPITT pixel bias and the number of outliers in the dataset. The sensitivity of each MOPITT pixel and each product is examined over the Canadian high Arctic. We then follow the methodologies recommended by NDACC and TCCON for the comparison between the FTIR and satellite total column retrievals. MOPITT averaging kernels are used to weight the NDACC and TCCON retrievals and take into account

the different vertical sensitivities between the satellite and PEARL FTIR measurements. We use a modified Taylor diagram to present the comparison results from each pixel for each product over land and water with NDACC and TCCON measurements. Our results show overall consistency between MOPITT and the NDACC and TCCON measurements. When compared to the FTIR, the NIR MOPITT retrievals have a positive bias of 3-10% depending on the pixel. The bias values are negative for the TIR product, with values between $-5\%$ and $0\%$. The joint TIR-NIR products show differences of $-4\%$ to $7\%$. The

drift in MOPITT biases (in units of $\%\,\mathrm{year}^{-1}$) relative to NDACC and TCCON varies by MOPITT data product. In the NIR, drifts versus TCCON are smaller than those versus NDACC, however, this scenario is reversed for the MOPITT TIR and joint TIR-NIR products.

## 1 Introduction

Carbon monoxide (CO) is a trace gas that has an important role in air quality, climate and atmospheric chemistry. It is produced

at the Earth's surface by incomplete combustion processes such as the burning of fossil fuels, biomass burning, and wildfires. The largest concentrations of CO in the lower and middle atmosphere are in the troposphere. Its chemical lifetime in the troposphere is on the order of weeks to a few months and this is dominated by the reaction with hydroxyl (OH) radical. Because of its long lifetime, CO can be used to trace pollution sources, and its global distribution provides information on the transport paths of pollution (Crutzen and Andreae, 1990). CO also has an important role in the atmospheric OH budget as

the main sink of CO is oxidation by OH. CO influences climate change indirectly by affecting greenhouse gas concentrations through producing carbon dioxide and tropospheric ozone via CO oxidation and limiting OH levels, thus increasing methane concentrations (Seinfeld and Pandis, 2006).

The Arctic is very sensitive to the Earth's climate and it is the place that is most affected by climate change (ACIA, 140pp, 2004). Pollutants and wildfires originating from North America, Europe, and Asia all contribute to Arctic pollution. North

America is the largest contributor for ozone, Europe for CO and aerosols at the Arctic surface, and East Asia for aerosols at higher altitudes in the Arctic (Shindell et al., 2008). This pollution affects the climate in the Arctic by changing the radiative budget as well as increasing summer sea-ice melt from the deposition of black carbon on snow and ice (Law and Stohl, 2007). Tropospheric ozone has an impact on the Arctic winter and spring warming. Tropospheric ozone can be produced photochemically at mid-latitudes or during transport to the Arctic (Pommier et al., 2010). CO is one of the precursors of ozone

through photochemical production. Global warming has various sources, one of them, directly and indirectly, is CO. Climate change has caused Arctic temperatures to rise significantly during the last few decades. The Canadian Arctic in particular has experienced $2.3\,°\mathrm{C}$ warming, three times the global mean warming rate (Bush and Lemmen, 2019), since ground-based measurements by the Joint Arctic Weather Stations (JAWS) program began in 1947.





Nadir-viewing satellite instruments can provide a global view of atmospheric composition with dense geographical coverage. Over the past two and a half decades, CO has been measured from space using a suite of nadir sounders. One of the earliest satellite-based instruments that measured CO was the Interferometric Monitor for Greenhouse Gases (IMG) (1996) (Wang et al., 1998) which collected eight months of data in 1996-1997. This was followed by the launch of Measurements of Pollution

in the Troposphere (MOPITT) (Drummond and Mand, 1996) in 1999, the Scanning Imaging Absorption Spectrometer for Atmospheric CHartographY (SCIAMACHY) operating from 2002-2012 (Bovensmann et al., 1999), the Atmospheric InfraRed Sounder (AIRS) launched in 2002 (Aumann et al., 2003), and the Tropospheric Emission Spectrometer (TES) measurements from 2004 to 2018 (Beer, 2006). More recently, CO measurements are being made as part of operational missions including the Infrared Atmospheric Sounding Interferometer (IASI-A, -B, and -C) launched in 2006, 2012, and 2018, respectively (Clerbaux

et al., 2009), the Cross-track Infrared Sounder (CrIS) on Suomi NPP and JPSS-1 launched in 2011 and 2018 (Han et al., 2015), and the Tropospheric Monitoring Instrument (TROPOMI) launched in 2017 (Veefkind et al., 2012). The most recent mission in this suite is the Greenhouse gases Observing SATellite-2 (GOSAT-2) that was launched in 2018 (Suto et al., 2020). MOPITT measures CO in both the thermal infrared (TIR) and near infrared (NIR) spectral regions. For comparison, AIRS, TES, IASI and CrIS measure CO spectra in the TIR, and SCIAMACHY, TROPOMI and GOSAT-2 operate in the NIR. To date, the longest

record of global CO measurements is provided by MOPITT, at more than 20 years.

In order to use satellite instrument time series for climate and air quality studies, it is important to validate the dataset. Therefore, comparisons between satellite data and long-term aircraft- and ground-based measurements are essential. Ground-based Fourier Transform Infrared spectrometers (FTIR) measure the solar radiation that has passed through the atmosphere, which can be analyzed to determine the total column of CO with high accuracy and precision. The two global ground-based

FTIR networks providing total column CO measurements are the Network for the Detection of Atmospheric Composition Change (NDACC) (De Mazière et al., 2018) and the Total Carbon Column Observing Network (TCCON) (Wunch et al., 2011a). NDACC measures CO in the TIR and TCCON in the NIR. NDACC and TCCON measurements of the column-averaged dry-air mole fraction of CO, $X_{CO}$, have been compared in recent studies (e.g., Kiel et al., 2016; Zhou et al., 2019) Using simultaneous measurements taken over Karlsruhe, Kiel et al. (2016) found a 4.76% relative bias with a standard deviation

of 2.28% between NDACC and TCCON $X_{CO}$. Zhou et al. (2019) examined results from six NDACC and TCCON sites around the globe. They found that NDACC measurements were 5.5% larger than those from TCCON in the Northern hemisphere and that the difference between the two networks is within $\pm 2\%$ for the Southern hemisphere sites.

Several studies have used NDACC and TCCON data to validate recent MOPITT CO retrievals. Globally, Buchholz et al. (2017) used NDACC data from 14 sites to validate MOPITT version 6 (V6) and 31 TCCON sites were used by Hedelius et al.

(2019) to validate MOPITT version 7 (V7). These FTIR studies both found larger biases in the Arctic region. Buchholz et al. (2017) recommended not using MOPITT V6 data for trend analyses above 60 °N due to larger drift biases and Hedelius et al. (2019) showed a lower bound pixel-to-pixel bias of order 10 ppb over the Arctic for the MOPITT V7 data. In addition to these FTIR studies, Deeter et al. (2014, 2017, 2019) used aircraft data to validate MOPITT V6, V7 and version 8 (V8), respectively.

To examine the latest MOPITT dataset (V8) at high latitudes, we focus on validation comparisons in the Canadian high

Arctic using both NDACC and TCCON data. The remaining sections of the paper are arranged as follows. First, in Sect. 2, we





describe the MOPITT instrument and its data products, investigate pixel-to-pixel biases, and apply filters to remove outliers for the comparisons. In Sects. 3 and 4, we describe the FTIR measurements and the NDACC and TCCON datasets and discuss the vertical sensitivity of each retrieval and its averaging kernels. The validation methodology, including coincidence criteria and the comparison approach are explained in Sects. 5. The results of the validation comparisons for MOPITT with NDACC

and TCCON are shown in Sect. 6, including comparisons with previous results. Finally, we summarize the results and make conclusions in Sect. 7. Buchholz et al. (2017) and Hedelius et al. (2019) are referenced several times in this paper, therefore we allocate the names "Buchholz2017" and "Hedelius2019" respectively to reference them.

## 2  MOPITT satellite instrument

MOPITT is on-board NASA'a Terra satellite, which was launched in December 1999 (Drummond et al., 2010). The Terra

satellite is in a sun-synchronous, near-polar orbit with an inclination angle of $98.4\,°N$ at ~705 km altitude with an equator overpass time at 10:30 AM (descending node). MOPITT is a nadir-viewing multi-channel TIR and NIR instrument with horizontal spatial resolution of $22\times22$ km and a swath width of ~640 km which is achieved by cross-track scanning (Drummond and Mand, 1996; Drummond et al., 2010). This provides near-global measurement coverage from $82\,°N$ to $82\,°S$ in ~3 days. MOPITT uses a correlation spectroscopy technique, employing pressure- and length-modulated gas cells, to measure CO

concentrations. Although the instrument comprised eight channels originally, only three channels have been used to retrieve CO since August 2001 due to a failure in the cooling system. Of these channels, two are in the TIR band ($4.6\,\mu$m) and one is in the NIR band ($2.3\,\mu$m). The TIR channels have the most sensitivity to middle and upper tropospheric layers and show significant sensitivity to CO variation, thus providing profile information, while the reflected solar (NIR) channels are sensitive to the total CO column. There is significant measurement sensitivity in the lower troposphere, if the temperature contrast

between the surface and the atmosphere is large (Drummond et al., 2010).

The MOPITT retrieval process utilizes an iterative optimal estimation method in log(volume mixing ratio (VMR)) to combine measured radiances and *a priori* information (Deeter et al., 2003). Each channel's detector is comprised of a four-pixel linear array, where 1 and 4 are the outer pixels and 2 and 3 are the inner pixels of the array. For each pixel, the retrieved profiles are provided on a 10-level fixed-pressure grid as the average VMR within each layer, where these levels correspond

to the pressure at the bottom of each layer (Deeter et al., 2013). These are also integrated to provide MOPITT total column CO values. In addition, for each pixel, the type of surface is catalogued as water, land, or mixed (coastline). We use V8 of the MOPITT level 2 data in this study including TIR-only, NIR-only, and joint TIR-NIR products (Deeter et al., 2019). The joint TIR-NIR retrievals use radiances from both channels and provide profiles with the largest degrees of freedom for signal (DOFS), the best vertical resolution, and the highest sensitivity in the lower troposphere (Deeter et al., 2015). We compared

with the TIR and joint TIR-NIR products from MOPITT over both land and water. MOPITT NIR (solar reflectance) retrievals provide information only over land.

Each MOPITT version product provides improvements over the previous version. As we will compare our results with some results from MOPITT V6 and V7, it is useful to briefly mention the improvements from V6 to V7 (Deeter et al., 2017)





and then from V7 to V8 (Deeter et al., 2019). The first improvement in V7 is the consideration of the steady growth in $N_2O$ concentrations in the atmosphere over time in the radiative transfer model, rather than using constant concentrations for this interfering species. This could produce a time-dependent bias in calculated radiances and possible retrieval drift. The second improvement is changing the source of the meteorological fields used (such as water vapour and temperature

profiles and surface temperature) from NASA MERRA (Modern-Era Retrospective Analysis for Research and Applications) reanalysis products for V6 to MERRA-2 for V7. The MOPITT retrieval algorithm only considers the observations of clear-sky as input, which is determined from MOPITT's thermal channel radiances and the MODIS (Moderate Resolution Imaging Spectroradiometer; also aboard Terra) cloud mask. For this cloud detection, MODIS Collection 6 was used for the V6 retrievals after March 2016, and was used for the entire MOPITT V7 dataset. This change mostly affects the number of clear-sky scenes

over the tropics specifically during nighttime. In the MOPITT retrieval process, the simulated radiances calculated by the operational radiative transfer model are compared to the actual calibrated level 1 radiances and the bias between them is corrected. Radiance-bias correction factors compensate for different bias sources like forward model errors due to assumed spectroscopic data, geophysical errors, and errors in instrumental specifications (Deeter et al., 2014). For V7, a new strategy was used to derive radiance-bias correction factors by minimizing observed retrieval biases at 400 and 800 hPa using in situ

CO profiles.

The most recent release, MOPITT V8, has several enhancements over V7. In V8, a new water vapor model and collisionally-induced nitrogen absorption have been implemented (Deeter et al., 2019). The second change is in the radiance-bias correction. The new parameterization includes the date and geographical location of the MOPITT observation and that water vapour total column at the observation time. This method decreases both retrieval drift and geographical variability of the biases. Another

improvement is in the cloud detection, where MODIS Collection 6.1 cloud mask is applied, and also the threshold ratio value of radiance for cloudiness is increased.

## 2.1 MOPITT pixel-to-pixel biases

There have been several studies that investigated pixel-to-pixel biases among the four MOPITT pixels. Deeter et al. (2015) found that the MOPITT V6 retrieval performance varies based on instrumental and geophysical effects and discussed how

filtering could be used to reduce the impact of variations in instrumental noise between pixels. Globally for MOPITT V7, Hedelius2019 investigated the pixel-to-pixel biases and their trends for snow- and ice-free pixels. They observed that pixel 1 had the largest negative bias and found that for all pixels the biases grow increasingly larger moving polewards, with pixel 2 having a smaller bias than pixel 4 at high latitudes. Also, Buchholz2017 examined how validation comparisons for V6 differed between pixels. They found that the correlations were poorer for pixel 1 than the other pixels for all data products and found

the best correlations across data products for pixel 3.

To examine the pixel-to-pixel biases for MOPITT V8 over the Canadian high Arctic, we calculated 30-day means of MOPITT total column CO measurements from the joint TIR-NIR retrieval within a radius of 110 km around Eureka, Nunavut for each pixel over land or water and then compared these to the weighted mean of the measurements from all pixels for the same time period. This internal comparison of MOPITT data quality is based on the assumption that each pixel samples the same





area. These pixel-to-pixel bias results are plotted in Fig. 1 with the average over the whole period plotted as a corresponding line. Pixel 1 over land has the largest negative bias, which is consistent with Hedelius2019. The pixel 1 bias is larger over land than over water. Pixel 3 has a large variability in bias over land although it oscillates around zero; therefore the average is small and positive. Pixel 2 has the smallest bias compared to other pixels. The variability in bias for pixel 4 is smaller than for pixel 1.

5    However, the average bias for pixel 4 over land is relatively large and positive. Overall, the variability in bias is more consistent for all pixels over water and the average biases are smaller than those over land. Hedelius2019 found a significant trend in the pixel-to-pixel bias over time, therefore, they applied a bias correction before validation. In our case, the variability of the pixel-to-pixel bias is large (due to our smaller statistics) and the average biases all fall within each others' standard deviations. Therefore, we did not make a bias correction.

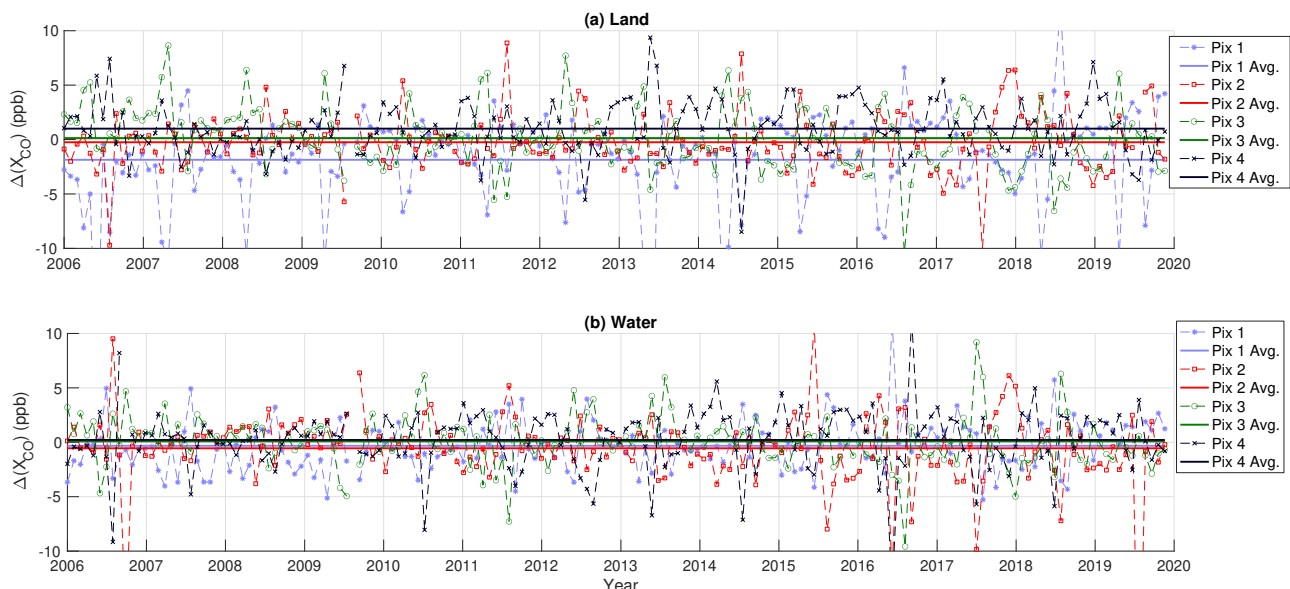

**Figure 1.** Individual MOPITT pixel biases, compared to the weighted mean of all pixels, over time for the joint TIR-NIR product. Symbols show 30-day mean biases and thick lines show all year averages for pixels 1 (blue), 2 (red), 3 (yellow) and 4 (black) over land (a) and water (b).





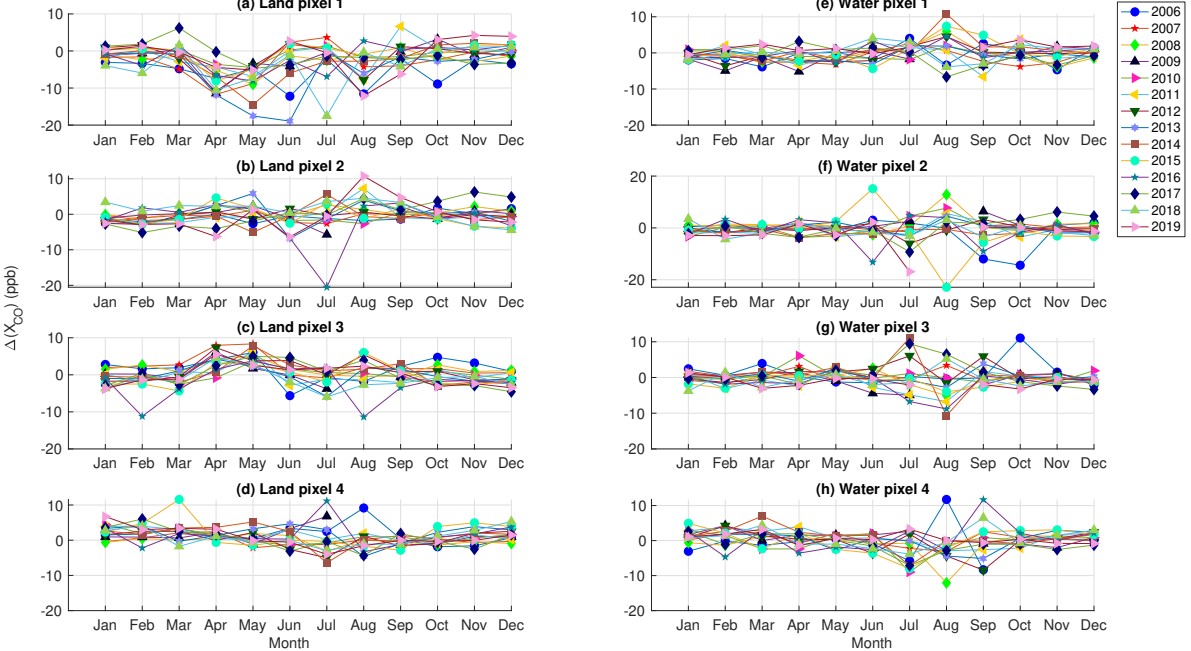

**Figure 2.** Monthly MOPITT pixel biases compared to the weighted mean of all pixels over land and water over each year. Colours and symbols indicate each year from 2006 to 2019.

The outliers and pixel biases in Fig. 1 appear to have some periodicity, so to examine, the oscillations for each pixel and investigate seasonal effects, the monthly average of the pixel bias for each year in the 110 km radius circle around Eureka is plotted in Fig. 2. Monthly snow and ice background percentage taken from MODIS (provided in the MOPITT data files), as well as solar zenith angles, are plotted in Fig. 3. Pixel 1 has a large negative bias over land in the spring and summer months.

5 The depth of snow over the Eureka region is at a maximum in spring and this could be the reason for the larger biases over land, which are shown in Fig. 3 of Howell et al. (2016). Over land, pixel 2 has almost no bias. Pixel 3 over land has little bias except during April and May when the bias is positive. Pixel 4 has a positive bias all year except during the summer when snow and ice background is minimal. We observed large biases for all pixels over water during July and August which is correlated with the minimum amount of ice during those months and minimum solar zenith angles. Most of the pixel biases are seen in

10 July-August when there is a mixture of ice and water over the ocean and the snow/ice background percentage over the ocean is at a minimum.





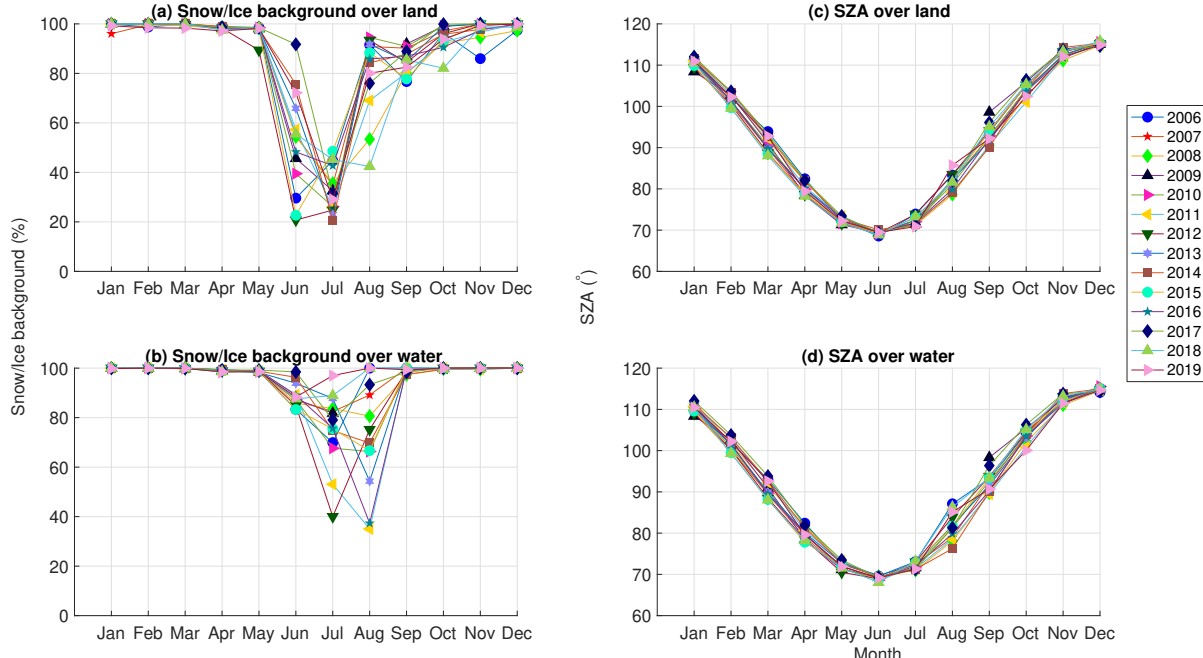

**Figure 3.** Monthly MODIS snow/ice background percentage over land (a) and water (b), and monthly solar zenith angle over (c) land and (d) water corresponding to MOPITT measurements shown in Fig. 2.



## 2.2 MOPITT filtering

We adopted the filtering method of Hedelius2019 for our study. This uses the "small-area approximation" or "small-region approximation" (known as SRA) to identify outliers in the dataset, based on the assumption that over a small enough area (1° radius), on a single orbit track, atmospheric properties are almost homogeneous (e.g., Mandrake et al. (2015); Wunch et al. (2017); O'Dell et al. (2018)). To do this, the median value of the MOPITT CO retrievals for each pixel is subtracted from all the retrievals from each pixel in the small region to calculate the anomalies. Figures 4 and 5 show the anomaly results plotted versus different parameters that may affect the retrieval, and include the histogram of the distribution of the measurements for each parameter, systematic biases from zero in the mean pixels, pixel bias for each pixel, and the root-mean-square (RMS) residual from the SRA for each parameter. These plots were used to determine the parameters to focus on and the filtering criteria to be used by examining the variation and spread in the pixel-to-pixel biases. Table 1 summarizes the filter parameters determined and the limits we apply to the MOPITT data to minimize the outliers, separated for land and water. Also, the percentage of MOPITT data that is passed by the filters is reported for each parameter. Most of the data removed are filtered due to the SZA parameter (~40% pass percentage) which was chosen to limit the data to daytime-only when the comparison FTIR is measuring and to be consistent with previous studies. Of the remaining filter parameters, those that have greatest impact are the MODIS snow/ice background and signal Chi-squared ($\chi^2$), which represents the goodness of the retrieval's fit (~90-93% and ~83-90 % pass percentage, respectively). Because low DOFS retrievals are associated with low CO concentrations and removing them would induce a positive bias in our comparisons, we did not use DOFS as a filter parameter following the recommendation of Deeter et al. (2015). Also, based on the MOPITT data product recommendations, we did not use the surface temperature as a filter parameter, because it is a physical parameter.

**Table 1.** Parameters used for filtering and evaluating the MOPITT data to minimize outliers in the dataset, and the percentage of retrievals that pass the specified threshold.

| Parameter | Limits-land | Pass percentage | Limits-water | Pass percentage |
|---|---|---|---|---|
| Solar zenith angle (SZA) | < 90 | 39.58 | < 90 | 40.99 |
| Degrees of freedom for signal (DOFS) | - | - | - | - |
| $\chi^2$ | < 10 | 89.77 | < 60 | 82.60 |
| Surface emissivity | > 0.76 | 99.82 | > 0.58 | 99.44 |
| Error in surface emissivity | >0.045 | 97.36 | >0.035 <0.057 | 98.96 |
| MODIS snow/ice background | < 0.1 | 89.04 | <0.1 | 93.10 |
| Elevation (m) | < 1000 | 93.78 | < 20 | 99.48 |
| Surface temperature | - | - | - | - |


**Figure 4.** SRA bias for the parameters that affect the MOPITT CO retrievals over land. The panels show solar zenith angle (SZA) (a), degrees of freedom (b), Chi squared (c), Surface emissivity (d), error of surface emissivity (e), snow/ice background (f), elevation (g), and surface temperature (h). The blue stars show the overall mean bias of all pixels, the dark green, red, light green and yellow stars are for pixel 1 through 4, respectively. The normalized histogram of the spread of the data are grey with the corresponding axis on the right. The RMS from the SRA are shown by the blue circles.





**Figure 5.** Same as Fig. 4 but for retrievals over water.



## 3 FTIR instrument

Ground-based high-spectral-resolution FTIRs operating in transmission mode are widely used to measure atmospheric trace gases, including CO. The atmospheric absorption spectra produced by these instruments are used to retrieve total and partial column densities by exploiting atmospheric pressure broadening (Pougatchev et al., 1995). There are two global networks spanning from the Arctic to the Antarctic that utilize these instruments to study the Earth's atmosphere, namely NDACC (De Mazière et al., 2018) and TCCON (Wunch et al., 2011a). Since 2006, a Bruker IFS 125HR FTIR has been operating at the PEARL Ridge Laboratory in Eureka, Nunavut, Canada (80.05° N and 86.42° W, 610 m a.s.l.; Fogal et al. (2013)) and currently contributes to both measurement networks. This instrument operates during clear-sky conditions from polar sunrise (~February 20) until polar sunset (~October 20) and typically records infrared solar absorption spectra on 80-120 days per year (Strong et al., 2017). From here onward, the Eureka FTIR measurements used in this study will be referred to by the network name (e.g., NDACC or TCCON).

### 3.1 NDACC

The NDACC FTIR spectral coverage is obtained using two detectors (InSb and HgCdTe) which cover the MIR from 600 to 4800 cm$^{-1}$. The instrument is operated at a spectral resolution of 0.004 cm$^{-1}$ (unapodized). VMR profiles are retrieved from the FTIR spectra and total and partial column densities are determined by converting VMR to density using temperature and pressure profiles (Batchelor et al., 2009). SFIT4, a profile retrieval algorithm based on the optimal estimation method (Rodgers, 2000), is used with a combination of *a priori* information and information in the recorded spectral measurements to perform the spectral fitting. In the optimal estimation method, the VMR profile is iteratively updated until the difference between the measured and calculated spectra is minimized. The mean outputs from Whole Atmosphere Chemistry Climate Model (WACCM) version 4 between 1980-2020 are used for the *a priori* VMR profiles (Marsh et al., 2013) and daily temperature and pressure profiles from the National Centers for Environmental Protection (NCEP) interpolated to the geographical location of PEARL are used in the retrieval (ftp://ftp.cpc.ncep.noaa.gov/ndacc/ncep/). Version v0.9.4.4 of the SFIT4 retrieval software is used here (https://wiki.ucar.edu/display/sfit4/). The NDACC FTIR dataset provides CO partial columns in units of molecules cm$^{-2}$ as well as vertical profiles of CO in VMR, on a fixed altitude grid with 47 levels between 0.8 km and 113.0 km (with these altitudes corresponding to the centers of the retrieval layers). Version 5 of the NDACC data is used in this study for the period between August 2006 to October 2019. The number of measurements during 2012 and 2013 is less than in other years due to operational limitations.

### 3.2 TCCON

The TCCON FTIR spectra are measured in the NIR from 3800 to 11000 cm$^{-1}$ at a spectral resolution of 0.02 cm$^{-1}$ using an InGaAs detector. Estimates of column-averaged dry-air mole fractions ($X_{CO}$) are retrieved from the measurements; therefore we use TCCON measurements to compare with the CO total column MOPITT values. The GFIT spectral fitting algorithm is used to retrieve trace gas amounts. It uses a nonlinear least squares spectral fitting algorithm that scales the *a priori* profile to





produce a calculated spectrum that best matches the measured spectrum (Wunch et al., 2011a). The algorithm integrates the scaled profile to calculate the column abundance and the dry-air mole fractions are then calculated by dividing the column abundance by the column of dry air obtained from the simultaneous $O_2$ column abundance measurement. The TCCON CO *a priori* profiles are based on an empirical model and the temperature, pressure, and humidity *a priori* profiles are based on

NCEP/National Center for Atmospheric Research reanalyses (Wunch et al., 2011a). TCCON $X_{CO}$ is reported in units of ppb. The TCCON data used for this study are version GGG2020 (Laughner and the TCCON team., 2020) from March 2010 to October 2019.

## 4 Vertical sensitivity of instruments

In order to compare the MOPITT CO measurements with those from NDACC and TCCON, the vertical sensitivity of each

instrument must be taken into account. Figures 6(a-c) show the MOPITT averaging kernel (AK) rows for the all-pixel average (2006-2019) observed within a 110 km radius from Eureka for the NIR-only, TIR-only, and joint (TIR-NIR) CO retrievals, respectively. The greatest sensitivity in all three products is in the upper troposphere with the maximum around 400 hPa. The advantage of the multispectral joint TIR-NIR product over the single channel (TIR-only or NIR-only) products is clear in the improvements seen near the surface (900 hPa) and in the upper troposphere. The MOPITT total column AKs corresponding

to Fig. 6(a-c) are presented in Fig. 6(e-g) and are separated by pixel over land and water. For the total column NIR-only AK, the sensitivity is higher in the upper troposphere and pixels 1 and 4 show the maximum and minimum sensitivity, respectively. However, the difference between the pixels is not large. The total column TIR-only AK for all pixels over water is approximately twice as large as that over land, and the difference between them is noticeable. The reason could be due to the larger geophysical noise over land than over water. The geophysical noise is affected by surface height and emissivity (Deeter et al.,

2011). The variation of elevation over land around Eureka (Fig. 4(g) and Fig. 5(g)) is large, ranging from sea level up to approximately 1,500 m. Also, the surface emissivity over land (Fig. 4(d) is much larger than over water (Fig. 5(d)). TIR averaging kernels are also dependent on the temperature difference between the surface and the air above it (thermal contrast); the effect on the averaging kernels due to changes in thermal contrast also has a seasonal component. Over the Arctic, the thermal contrast over water is smaller than over land (Fig. 3(a) and (b)) as the snow and ice background percentage are larger over water

than land. Therefore, the averaging kernels over water are more sensitive to the free troposphere while the averaging kernels over land show some sensitivity to the lower troposphere. For these measurements, Pixel 1 has the maximum sensitivity, and Pixel 3 has the minimum. The differences between land and water measurements decrease for the total column joint TIR-NIR products and their AKs are more similar than for the TIR-only retrievals. Overall, pixel 1 has the highest sensitivity and pixel 3 shows the lowest sensitivity in the joint TIR-NIR products. The contribution of NIR measurements to the joint retrievals

improves the sensitivity of the AKs in the lower troposphere. Calculated from the trace of the AK matrix, the DOFS represents the information content of the retrievals. The monthly average DOFS for each MOPITT product by pixel and surface are presented in Fig. 7(a-c). The variation of the DOFS over the year and between pixels can be seen in each plot. MOPITT AKs vary with season, which is reflected in the DOFS seasonal variability. The DOFS for the joint TIR-NIR product are higher than that





for the TIR-only and NIR-only products, the land DOFS are larger than those over water, and pixels 1 and 3 have the largest and smallest DOFS, respectively (Fig. 7(a-c)). Figure 7(d) shows the monthly average DOFS for the NDACC CO retrievals by year. There is variation of the DOFS over the year typically with SZA, with lower DOFS in summer when the Sun is highest in the sky. The DOFS for the NDACC retrievals which are roughly twice those of MOPITT. The average DOFS for the NDACC

measurements is around 2 and that for the MOPITT joint TIR-NIR product is around 1.

Figures 6(d) (dashed line) and 6(h) demonstrate the total column AKs for the NDACC and TCCON CO measurements at Eureka, averaged over 2006-2019 and 2010-2019 respectively. It is necessary to mention that Figs. 6(d) and 6(h) are only shown up to 100 hPa to be comparable with MOPITT AKs. The TCCON AK varies weakly with SZA, with a maximum spread of around 0.1 at the surface. The TCCON total column AKs are less than unity below 400 hPa and they are above unity from

10 400 hPa to higher altitudes. This indicates that there is more sensitivity to the upper troposphere and above. In the case of NDACC, the total column AK being close to one indicates that all altitudes contribute to the total column equally. In contrast to the MOPITT total column AK, both NDACC and TCCON total column AKs are closer to unity than MOPITT indicating a larger contribution of the measurements to the retrieval rather than *a priori* information for these Eureka ground-based measurements. Because of these differences, the NDACC and TCCON retrievals will be smoothed by the MOPITT AKs for

our comparisons. Rodgers and Connor (2003) presented a general method for comparing measurements from two instruments with different averaging kernels, by smoothing the retrievals of the instrument with higher DOFS (higher-resolution) with the averaging kernels of the lower-resolution instrument. The details of the intercomparison methodology in this study are described in the next section.





**Figure 6.** Mean MOPITT retrieval averaging kernels between 2010 and 2019 within a 110 km radius from Eureka over land for all pixels: (a) NIR only, (b) TIR only, and (c) multispectral TIR + NIR. MOPITT total column averaging kernel for each pixel over land and water for (e) NIR only, (f) TIR only, and (g) multispectral TIR + NIR. NDACC FTIR averaging kernels and TCCON FTIR column averaging kernels for Eureka, averaged over 2006-2019 and 2010-2019 are in (d) and (h), respectively.



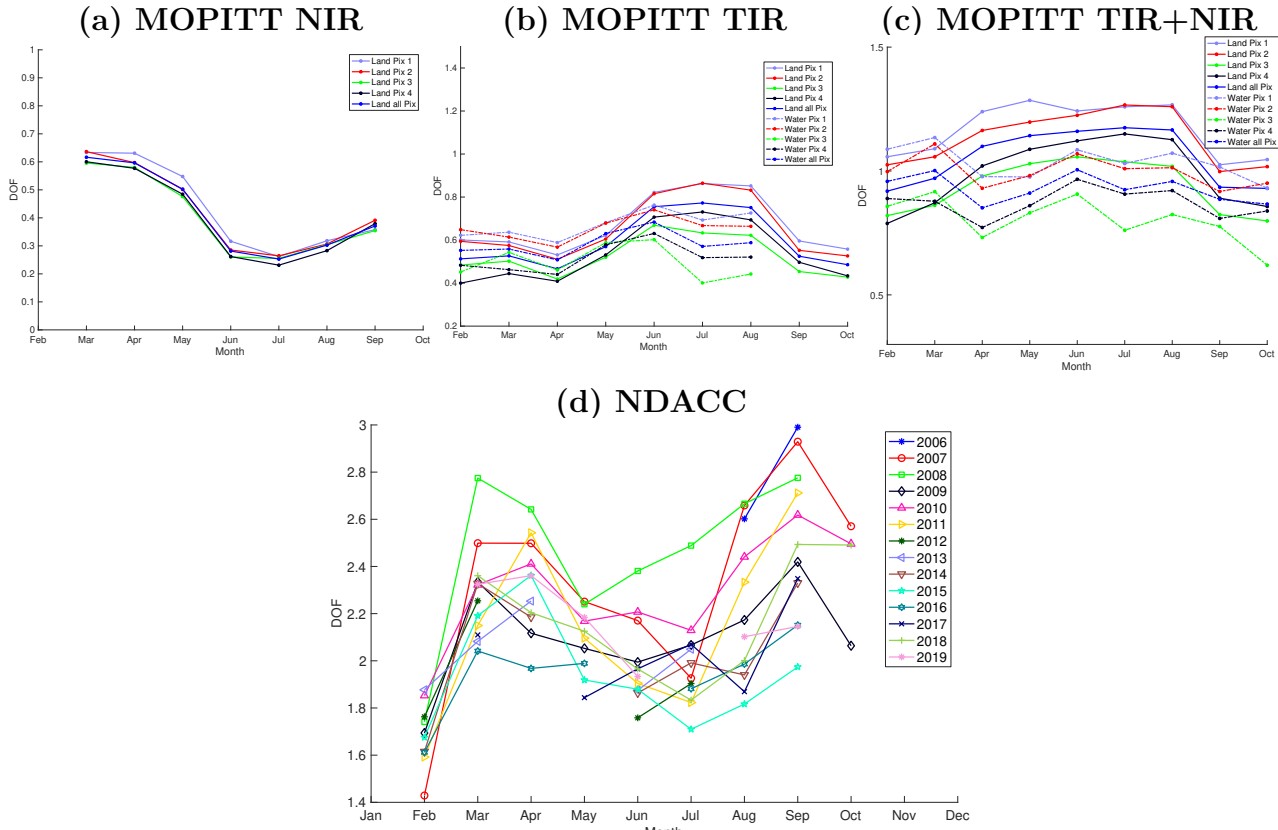

**Figure 7.** MOPITT monthly average DOFS from 2006 to 2019 for (a) NIR only, (b) TIR only, and (c) multispectral joint TIR-NIR. (d) NDACC FTIR monthly DOFS shown by year.

## 5 Data processing and validation methodology

### 5.1 Coincidence criteria

The coincidence criteria used in this work are consistent with the previous study by Buchholz2017 using NDACC measurements. MOPITT measurements are limited to be within the same day (24 h) as each FTIR measurement and only daytime measurements (SZA < 90°) are used. The MOPITT measurements must be within a 110 km radius from the PEARL Ridge Laboratory. This is a tighter spatial criteria than that used by Hedelius2019 in their TCCON comparisons, who used an area of 2° × 4° as a coincidence criteria globally but an area of 4° × 8° for stations above 60 °N. Figure 8(a) shows the location of PEARL at Eureka in the Canadian high Arctic. The topography of the area around Eureka is displayed in Fig. 8(b). There is a





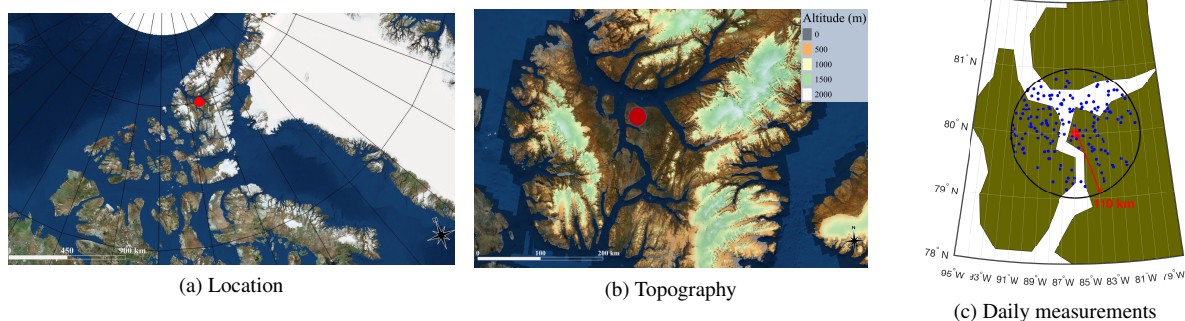

(a) Location        (b) Topography       (c) Daily measurements

**Figure 8.** (a) Geographic map showing location of PEARL at Eureka, Nunavut, Canada. (b) Topography around PEARL. (c) Example of daily MOPITT CO measurements near Eureka on 20 July 2007 within 110 km radius. In all figures, the red dot indicates the location of the PEARL Ridge Laboratory.

large variation in the topography over a small area (~200 km radius) with a mixture of water and land in the vicinity of Eureka. The QGIS 3.1 software is used to plot the data in Fig. 8(a) and 8(b) using WGS84 / NSIDC Sea Ice Polar Stereographic North data from US National Snow and Ice Data Center (NIMA Technical Report 8350.2, 1984). Figure 8(c) shows a map of the collocations between the PEARL FTIR and MOPITT measurements within a 110 km radius in July. Each blue dot represents a
MOPITT measurement. The time period for this study is between August 2006 to September 2019 for NDACC comparisons, and between July 2010 and August 2019 for the TCCON comparisons, when measurements are available.

## 5.2  Methodology

Several steps were taken to prepare the MOPITT retrievals and FTIR measurements for the validation comparisons. First, the MOPITT retrievals were filtered based on the criteria in Table 1. Then, for each NDACC or TCCON measurement, we
selected all of the co-located MOPITT measurements within ±12 hours of the FTIR measurement. From that subset, we further separated the MOPITT data for each retrieval by pixel number and surface type and then took a weighted average of each pixel and land type subset of the MOPITT measurements to compare with the single FTIR measurement. The weighted average is calculated based on Eq. 1 using the inverse-squared retrieval standard deviations as weights.

$$\bar{X}_{CO} = \frac{\sum\limits_{i}^{N} X_i X_{i,\sigma}^{-2}}{\sum\limits_{i}^{N} X_{i,\sigma}^{-2}}, \tag{1}$$

where $X_i$ is each MOPITT measurement with a corresponding standard deviation of $X_{i,\sigma}$. This is done because the variability in Eureka's geography can influence retrievals. There are retrievals with large uncertainties and the weighted average reduces the effect of these outlier retrievals. We also calculated the weighted average of the MOPITT AKs and *a priori* profiles in this same manner. The MOPITT VMR values for each pressure level are reported at the bottom of each level, but the FTIR VMR measurements are assigned to the middle of each FTIR level. In addition, the FTIR retrieval grids are finer than the





MOPITT retrieval grid. Therefore, it is necessary to re-grid the FTIR measurements. To do this, we used similar technique (approximation method) to that presented in Buchholz2017 interpolating the FTIR profiles on a log-pressure grid to an ultrafine grid of 500 grid points per MOPITT layer (rather than 100 grid points used in Buchholz2017). Then the FTIR profiles were averaged over the same pressure range as the MOPITT retrieval levels. Next we examined the surface pressure difference

between the FTIR and MOPITT measurements arising due to topography. Buchholz2017 noted that the significant surface altitude/pressure differences found between measurements from MOPITT and those from NDACC stations at high altitudes or with highly variable terrain like Eureka can create additional biases in the total column comparison. Therefore, it is necessary to consider the difference in the surface pressure to compare total column values over the same air mass range. We adjusted the surface pressure using the method explained in Buchholz2017, which is based on the method of Kerzenmacher et al. (2012).

Two scenarios are possible in these surface adjustments. First, the surface pressure at the FTIR site is smaller than MOPITT; and in our case this is the most likely scenario because of the altitude of the PEARL Ridge Laboratory. In this case, the gap between the FTIR surface and MOPITT surface was filled with the FTIR *a priori* profile. If the difference between MOPITT and FTIR surface pressure was greater than a critical value of 80 hPa, we eliminated the MOPITT profile from the comparison. Buchholz2017 used 50 hPa as the critical value and we found this limited the number of MOPITT profiles in the comparisons.

The second scenario is when the surface pressure at the FTIR site is larger than MOPITT. The fine-grid layers below the MOPITT surface pressure level are then eliminated.

The next step was to smooth the FTIR retrievals with the MOPITT AKs since the FTIR retrievals have larger DOFS. Buchholz2017 (NDACC data) and Hedelius2019 (TCCON data) used different techniques to compare their data with the MOPITT data. In order to maintain consistency to compare our results, we used similar techniques which are briefly presented

here. V8 MOPITT retrievals provide both the total column averaging kernel, based on the method of Rodgers (2000), and the AK matrix, which shows the sensitivity of the retrieved total column to perturbations at each level of the MOPITT CO profile (Deeter et al., 2019).

To compare with the NDACC measurements, we used the MOPITT total column averaging kernel vector ($\mathbf{a_M}$) to smooth the FTIR NDACC profiles ($\boldsymbol{x_{NDACC}}$) using Eq. 2.

$$C_{Nsmoothed} = C_{Ma} + \sum_{j=1}^{10} \mathbf{a_{M,j}}(log_{10}(\boldsymbol{x_{NDACC}}) - log_{10}(\boldsymbol{x_{Ma}}))_j,$$   (2)

where $C_{Ma}$ is the *a priori* total column value corresponding to the MOPITT *a priori* profile ($\boldsymbol{x_{Ma}}$). For V8, like other versions, the MOPITT retrieval is in log space and the averaging kernel matrix and $\mathbf{a_M}$, which is the vector of derivatives of CO partial column values with respect to perturbations in log(VMR), must be applied to the log of the MOPITT profiles. The general relation to calculate the dry-air mole fraction, called $X_{CO}$, is based on the ratio of CO total column ($C_{CO}$) to the total column

of dry air ($C_{dryair}$) (Kiel et al., 2016).

$$X_{CO} = \frac{C_{CO}}{C_{dryair}}.$$   (3)





The MOPITT total column CO retrievals are in units of $molec.cm^{-2}$ and the MOPITT data product contains a model dry-air column. Using these, we can calculate the dry-air mole fraction, in unit of parts per billion (ppb) using Eq. 4:

$$X_{CO}(ppb) = \frac{CO\ column\ (molec.cm^{-2})}{model\ dry\ air\ column\ (molec.cm^{-2})} \times 10^9. \tag{4}$$

To compare the NDACC results with MOPITT in units of ppb, the result of Eq. 2 should be converted to ppb. For this, the

$C_{dryair}$ for the NDACC data can be calculated using parameters provided in the NDACC data files such as surface pressure ($P_0$), the gravitational acceleration ($g$), and total column of water vapour ($C_{H_2O}$) with Eq. 5:

$$C_{dryair} = \frac{P_0}{g \cdot m_{dryair}} - \frac{C_{H2O} \cdot m_{H2O}}{m_{dryair}}, \tag{5}$$

where $m_{dryair} = 28.964 \times 10^{-3}/N_A kg/molecule$ is the molecular mass of dry air, $m_{H2O} = 18.02 \times 10^{-3}/N_A kg/molecule$ is the molecular mass of water vapour and $N_A$ is Avogadro's constant.

Hedelius2019 described different methods for comparing MOPITT and TCCON measurements. We used their recommended method II for our comparison with TCCON data, which is also the method presented in Wunch et al. (2011b). Method IV in Hedelius2019 is similar to the method we used for the NDACC data as described above. According to Wunch et al. (2011a), $\boldsymbol{x}_T$ is the scaled *a priori* profile ($\boldsymbol{x}_{aT}$) using a scaling factor ($\gamma$). which is calculated as

$$\gamma_T = \frac{C_T}{C_{aT}}, \tag{6}$$

and $C_T$ and $C_{aT}$ are the total column dry-air mole fraction and its *a priori* value, respectively. For the MOPITT validation with TCCON measurements, we compare $C_{Tsmoothed}$ in Eq. 7 with $C'_M$ in Eq. 8.

$$C_{Tsmoothed} = C_{aT} + \sum_{j=1}^{10} \boldsymbol{a_{M,j}}(log_{10}(\boldsymbol{x_T}) - log_{10}(\boldsymbol{x_{aT}}))_j, \tag{7}$$

$$C'_M = C_M + C_{aT} - C_{Ma} + \sum_{j=1}^{10} \boldsymbol{a_{M,j}}(log_{10}(\boldsymbol{x_{Ma}}) - log_{10}(\boldsymbol{x_{aT}}))_j, \tag{8}$$

where $C_M$ is the MOPITT total column CO value corresponding to $\boldsymbol{x}_M$. TCCON retrieves the total column of $O_2$ ($C_{O_2}$) and $C_{dryair}$ can be calculated through Eq. 9 assuming the dry-air mole fraction of $O_2$ is 0.2095:

$$C_{dryair} = \frac{C_{O_2}}{0.2095}. \tag{9}$$

In the next section, the MOPITT column-averaged dry-air mole fraction XCO for each of the four MOPITT pixels, and each product, separated over land and water, are compared with the NDACC and TCCON total column values. There are 27 com-





parison sets between MOPITT and NDACC measurements considering the different combinations of MOPITT measurements. The same number of comparisons is conducted between the MOPITT and TCCON measurements. The 27 combinations of the four pixels over land and water (eight total), all pixel measurements combined ("Pixel all") over land and water separately (two total), and all pixel measurements combined over land and water together (one total) then three times for each product (NIR,

TIR, and Joint TIR-NIR). The NIR channel does not measure over water, therefore six comparisons are not made using the NIR channel which reduces the comparisons from 33 to 27.

## 5.3   Comparison approach

As described in Sect. 5.2, we compare 27 combinations for each FTIR measurement. To help visualize the results, we have used a Taylor diagram Rochford (2020), to summarize the comparisons between the MOPITT measurements and the FTIR

measurements. The Taylor diagram is useful for evaluating and assessing multiple properties of the comparisons for each MOPITT pixel and it has been used in different fields including atmospheric science (e.g., (Hegglin et al., 2010) and (Sharma et al., 2017)). In the Taylor diagram, MOPITT measurements are normalized to the FTIR measurements, to show how well each MOPITT measurement agrees with the NDACC or TCCON measurements. The Taylor diagram quantifies the relationship between each MOPITT and FTIR dataset in terms of the Pearson correlation coefficient ($R$), the centered root-mean-square

difference (CRMSE), and their standard deviations. Taylor (2001) found that there is a geometric connection between these parameters. The CRMSE is the mean-removed RMS difference and it is calculated using Eq. 10 (in units of $\mathrm{ppb}^2$):

$$CRMSE^2 = \frac{1}{N}\sum_{i=1}^{N}((M_i - \bar{M}) - (F_i - \bar{F}))^2, \tag{10}$$

where $M_i$ and $F_i$ represent MOPITT and FTIR measurements, respectively, and $N$ is the total number of measurements. $\bar{M}$ and $\bar{F}$ are the means of each dataset, respectively. These means are used to calculate the percent difference bias:

$$bias = 100\frac{(\bar{M} - \bar{F})}{\bar{F}}. \tag{11}$$

The relationship between CRMSE, the standard deviations of the MOPITT ($\sigma_M$) and FTIR ($\sigma_F$) data, and the correlation coefficient is given in Eq. 12 and shown in Figure 1 of Taylor (2001) geometrically based on the law of cosines:

$$CRMSE^2 = \sigma_M^2 + \sigma_F^2 - 2\sigma_M\sigma_F R. \tag{12}$$

Each point in a Taylor diagram can be characterized by a phase and an amplitude, which need to be determined. The correlation

coefficient and CRMSE are the quantities that measure how well measurements from each MOPITT pixel agree with the FTIR measurements in phase and amplitude, respectively. The correlation coefficient is the quantity that provides complementary statistical information quantifying the correspondence between the measurements associated with each MOPITT pixel and the FTIR measurements. These various quantities can be plotted on a polar graph, where the radial distance from the origin is



the standard deviation of the MOPITT measurements ($r$). The azimuthal angle on the polar graph is the correlation between the MOPITT measurements and the FTIR measurements or $\theta = arc\,cos(R)$ (Kärnä and Baptista, 2016). Then the CRMSE is the radial distance from the position of a pixel data point that matches exactly the FTIR measurements ($r = \sigma_F$, $\theta = 0$). As suggested in Taylor (2001), the statistics can be normalized by the standard deviation of the FTIR measurements. Equation 10

becomes dimensionless if we divide both side of the equation by $\sigma_F^2$. This new graph is called the modified Taylor diagram and in the normalized graph, the perfect MOPITT measurement would be positioned at ($r = 1, \theta = 0$). The advantage of this modified Taylor diagram is that we can compare different pixels with different standard deviations and its disadvantage is that this graph is based on centered measurements ($M_i - \bar{M}$) and therefore it does not show any pixel bias. There are a couple of approaches, such as Elvidge et al. (2014) and Kärnä and Baptista (2016), which take into this issue into account. We follow the

approach suggested by the latter paper in which they normalized the root-mean-square error (RMSE) (Eq. 13) by $\sigma_F^2$ and call it the normalized RMSE (Eq. 14). Therefore, if the RMSE is:

$$RMSE = \frac{1}{N}\sum_{i=1}^{N}(M_i - F_i)^2, \tag{13}$$

then the normalized RMSE would be:

$$Normalized\,RMSE = \frac{1}{\sigma_F^2}\frac{1}{N}\sum_{i=1}^{N}(M_i - F_i)^2. \tag{14}$$

Normalized RMSE always has non-negative values and smaller values indicate better agreement between the MOPITT dataset for a given pixel and the comparison dataset. Based on Eq. 13, the normalized RMSE will be zero for a pixel with measurements identical to the FTIR, and will be 1 for a pixel that has measurements equal to the mean of the FTIR measurements ($M_i = \bar{F}$). Normalized RMSE is sensitive to outliers and values greater than 1 represent poor agreement between the MOPITT data and the FTIR data. The $R$ (Pearson correlation coefficient) values in the Taylor diagrams were calculated using ordinary least squares.

One point that should be considered is that $\sigma_F$ is the standard deviation of the FTIR measurements and it is not calculated from random or systematic uncertainties reported by each instrument. The FTIR standard deviations ($\sigma_F$) for the NDACC and TCCON measurements are calculated from Eq. 2 and Eq. 7, respectively. In the comparison with NDACC measurements, we calculate $\sigma_M$ using the standard deviation of the MOPITT measurements. In the comparison with TCCON measurements, we use the standard deviation of the result of Eq. 8 to calculate $\sigma_M$.

The drift of the MOPITT-NDACC and MOPITT-TCCON biases for each pixel was calculated by conducting a linear fit with respect to time. A bi-square weighted robust fitting method was used to perform the linear fit (Holland and Welsch, 1977) and the significance of the drifts was computed using a Student's t-test. The advantage of this fitting method over the ordinary least squares fitting is that it is less sensitive to data gaps and outliers and it has been used in other studies such as Adams et al. (2014) and Bognar et al. (2019).




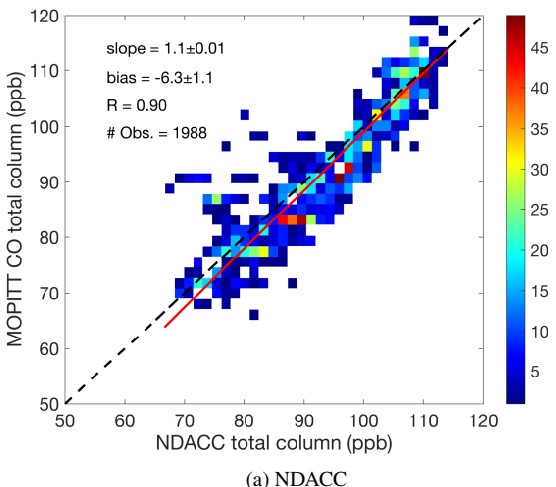

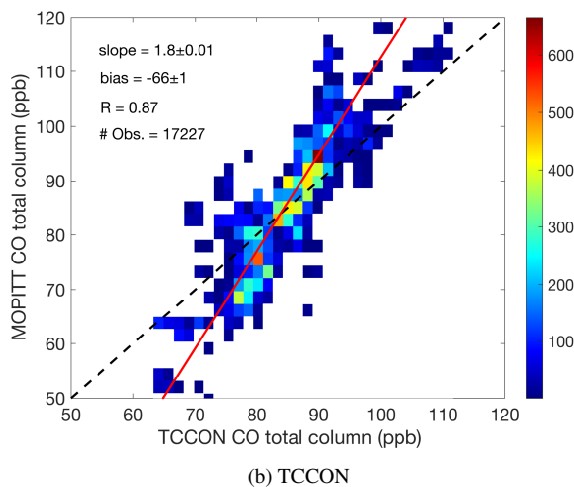

(a) NDACC

(b) TCCON

**Figure 9.** Sample correlation plots for MOPITT TIR pixel 2 CO measurements over land versus the (a) NDACC and (b) TCCON measurements using bi-square weighted robust fitting. The colors indicate the number of points in each bin to represent the density of points. Dashed black lines are 1:1 reference lines with slope of 1. Red lines are lines of linear best fit.

## 6   Results and discussion

As mentioned in Sect. 5.2, there are 27 comparisons between MOPITT CO total columns and each of the NDACC and TCCON datasets. The results for one of these comparisons are plotted in Figs. 9 (correlations) and 10 (time series of differences). These figures show results for the TIR pixel 2 comparisons over land. For this comparison, it can be seen that MOPITT has a
larger correlation coefficient and smaller bias drift with NDACC than with TCCON. Correlation and drift plots for all other comparisons are provided in the supplementary material. In the next subsections, the results of all 27 comparison sets for each pair of instruments are presented in Taylor diagrams.

### 6.1   Comparison with NDACC

The results of the comparisons between the MOPITT and NDACC column CO measurements are separated into measurements
over land, water, or both, and are shown in Fig. 11. In Fig. 11, column (a) shows the results for the MOPITT NIR product, column (b) shows the results for the TIR product, and column (c) shows the results for the joint TIR-NIR product. The first row of each column is a modified Taylor diagram, as described in Sect. 5.3, to provide a visual interpretation of the results to evaluate each pixel. The middle row represents the bias of each pixel versus normalized RMSE, and the bottom row shows the drifts for each pixel.
The Taylor diagrams for the NIR product show that all pixels have correlation coefficients between 0.7 and 0.8. Pixel 1 over land has a larger normalized standard deviation (NSTD) and normalized CRMSE than the other pixels. The Taylor diagram results for the TIR product reveal that the correlation coefficients for all pixels over land (~0.94) are larger than those over water (~0.85). Also, the same pattern can be observed for the normalized CRMSE, where all pixels over land are closer to

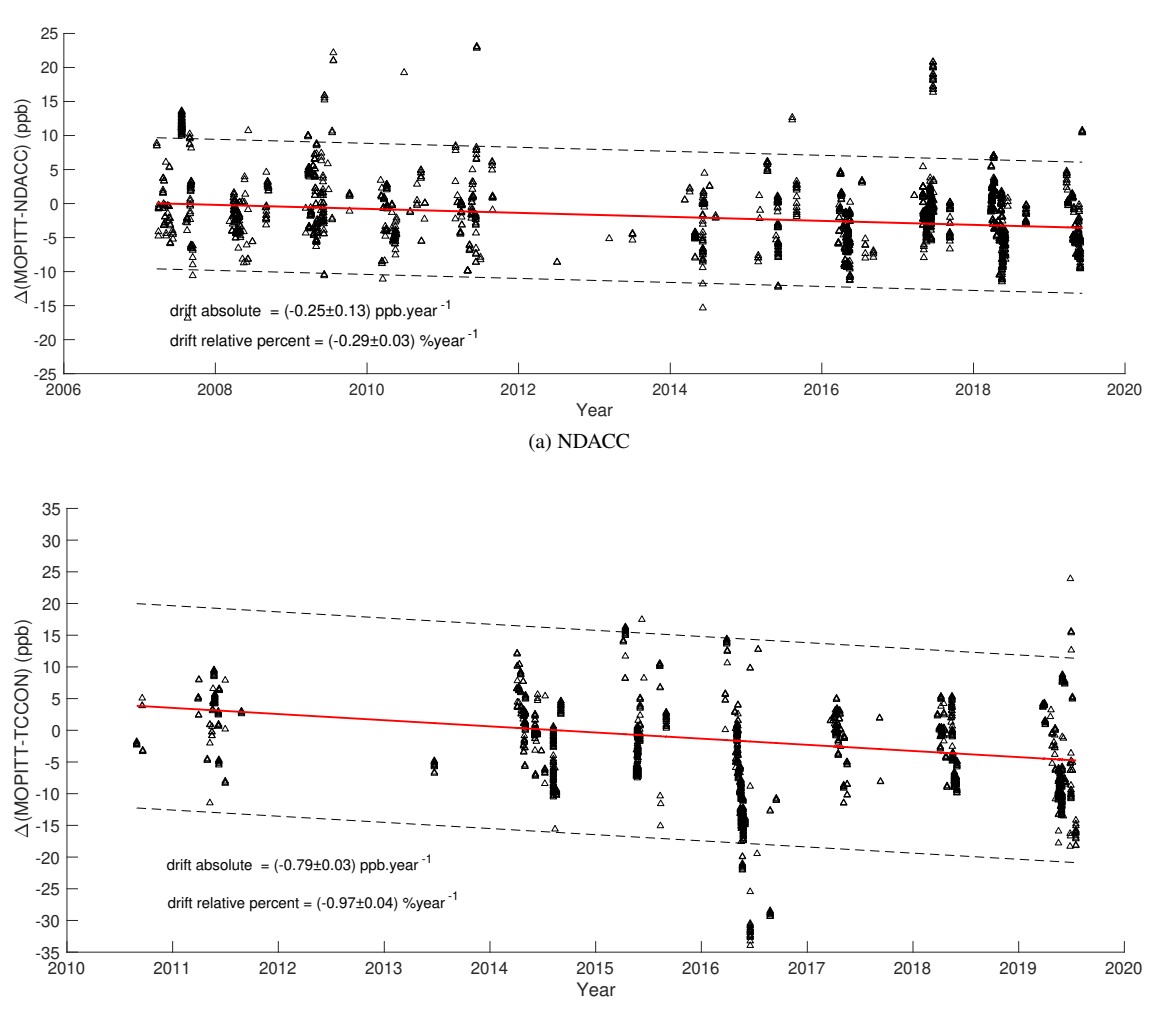

(a) NDACC

(b) TCCON

**Figure 10.** Sample temporal evolution of bias in MOPITT CO total column relative to (a) NDACC and (b) TCCON corresponding to Fig. 9. The red lines indicate bias drift calculated from a bi-square weighted robust fitting method. The dashed black lines are two sigma uncertainties.





the reference point (within 0.4 on CRMSE radial axes) versus pixels over water with values closer to 0.6. However, pixel 2 over land is the exception and its CRMSE and correlation values are worse than the other pixels over land. It is noted that the correlation coefficients and CRMSE for the TIR and joint TIR-NIR products for the all pixels combined (both land and water shown with the pink star) are almost identical to those for all pixels combined over land (shown as the purple square). The

results for the joint TIR-NIR product illustrate that the result of the all pixels combined over water is close to the all pixels combined over land. The best pixels for correlation coefficients for the joint product are the combined pixels over water and land (pink star) as well as pixel 3 over land (green circle). The overall NSTDs of the joint TIR-NIR pixels are larger than the NSTDs of the TIR measurements and they have slightly smaller correlation coefficients. Pixel 2 over water (pink triangle) has the smallest correlation coefficient in both the TIR and joint TIR-NIR products. Generally, the correlation coefficients found

for pixels over land are higher than those for pixels over water, which could be because of higher thermal contrast over land than water. Also, the correlation coefficients of the TIR products are larger than those of the joint TIR-NIR and both are larger than those of the NIR products. Similarly, the CRMSE values for the pixels over land are smaller than for pixels over water and they are closer to the reference point. In addition, the NSTD values of pixels over land are smaller than those over water with a few exceptions such as pixel 1 over land in the joint TIR-NIR products.

15       The second row of Fig. 11 illustrates the average bias versus RMSE for each pixel in Fig. 11. The error bars are the standard deviations of the bias values. The bias for the NIR shows that, on average, pixel 3 measurements over land have the smallest bias among all pixels over land, however, pixel 2's average bias is close to that of pixel 3. Pixels 1 and 4 have the largest bias and the normalized RMSE values are larger than 1. Also, all the pixels have a positive bias, therefore the MOPITT NIR measurements are generally larger than NDACC measurements, on average by roughly 5%. All pixel biases for the TIR product

are negative. The measurements for pixels over land have a lower bias and normalized RMSE than the pixel measurements over water. Pixels 1 and 2 over land have smaller biases than the others, followed by the combined pixel measurements over land and the combined pixel measurements over land and water. The joint TIR-NIR product biases are split with generally positive biases for pixels over land and negative biases over water, with the exception of pixel 1 over water which is close to zero and positive. The smallest bias and normalized RMSE are found for pixel 3 over land followed by pixel 1 over water. The bias of

pixel 1 over land is much larger than all the other pixels. Overall average pixel biases for all products agree within their standard deviations. Broadly, pixel biases over land are smaller than over water for the TIR products, and they are comparable in the joint TIR-NIR products except pixel 1 over land. Pixels 1 and 4 show large biases for the NIR products and their normalized RMSE is above 1.

The same overall pattern as found for the bias can be seen in the drift, in the third row of Fig. 11. All drift values for the

NIR are positive, with the largest values for pixels 1 and 4 and the smallest values for pixels 2 and 3 over land. The NIR pixel 1 drift uncertainty is larger than that of the other pixels. For the TIR products, the drift values for all pixels are negative and the drift values over land are smaller than those over water. Moving to the joint TIR-NIR product, the drift values over land are positive and then become negative over water, with the exception of pixel 2 over land, which has a negative drift. Pixel 1 over land has the largest drift value. The combined pixel measurements over land and the combined pixel measurements over land and water have close to zero drift. The drifts for all pixels have significance levels of 95% using the Student's t-test. The





pixels with significance levels of 95% are labeled with an asterisk (∗) in Fig. 11. The drift uncertainties of pixels over water are greater than those for the pixels over land.

Overall, the MOPITT NIR products show poorer performance than the TIR and joint TIR-NIR products. Pixel 1 over land shows a larger bias than the other pixels for the NIR and joint TIR-NIR products.

## 6.2 Comparison with TCCON

Figure 12 shows the comparison between MOPITT and TCCON measurements and is identical in format to Fig. 11. Here, we investigate each row of panels in the same order as above. In the modified Taylor diagrams, the correlation of coefficient for all pixels for all products is between 0.8 and 0.95. The NSTD values for the NIR product comparisons are around 1.6, except pixel 1 which is around 1.8. The NSTD values are between 1.5 and 1.8 for the TIR product and these increase to between 1.8 and 2.3 for the joint TIR-NIR product with a higher NSTD value of 2.5 for pixel 1 over land.

The normalized CRMSE for all pixels in the NIR product is around the 0.8 contour, with the largest value of 1.1 for pixel 1 over land. For the TIR product, the normalized CRMSE increases to values between 0.8 and 1 with the smallest value of 0.7 for the pixel 4 over land. The values increase significantly for the joint TIR-NIR product to between 1.1 and 1.5 with the largest value of 1.7 for pixel 1 over land. The normalized CMRSE values for joint TIR-NIR products are greater than those for the NIR and TIR products, which have a similar performance. In the Taylor diagram, results for almost all of the pixels tend to cluster in the same area, except for pixel 1 over land for the NIR and joint TIR-NIR products and for pixel 4 over land for the TIR product.

In the NIR, the pixel 3 has the smallest average pixel bias and pixel 1 has the largest. The normalized RMSE is around 1 for all pixels except for pixels 1 and 2, which have a larger bias than the others, with values close to 1.8 and 1.4, respectively. The TIR bias in the middle row shows that all pixel measurements over land cluster around zero percent bias and all pixels over water cluster around −3%. However, the normalized RMSE of all pixels over water is around 1, but the values for the pixels over land are less than 1. The joint TIR-NIR bias illustrates that all pixels have a normalized RMSE above 1. Pixel 1 has the highest bias at around 9% in the joint TIR-NIR product. For all comparisons, the pixel biases fall within each other's standard deviations due to the large scatter in the biases.

The drift values for the pixels over land for the NIR product are between $-1.3$ and $-1.9\,\%\mathrm{year}^{-1}$. Overall, the magnitude of the drift for all pixels for the NIR product is smaller than for other products, however, the TIR drift values for pixels over land are similar to those for the NIR product. For the TIR product, most of the pixels' drifts over land and water vary between $-1.0$ and $-2.0\,\%\mathrm{year}^{-1}$ except pixel 1 over water with $-2.8\,\%\mathrm{year}^{-1}$. For the joint TIR-NIR products, the drift tends to be worse than the drifts of the NIR and TIR products. The drifts for the joint TIR-NIR products are approximately twice as large, spanning $-2.5$ to $-4.5\%\mathrm{year}^{-1}$ except for pixel 3 over water, which has a smaller value than the others ($-1.5\%\mathrm{year}^{-1}$). Note that, the drift uncertainties are plotted for each pixel but they are not always visible because of their small magnitudes.

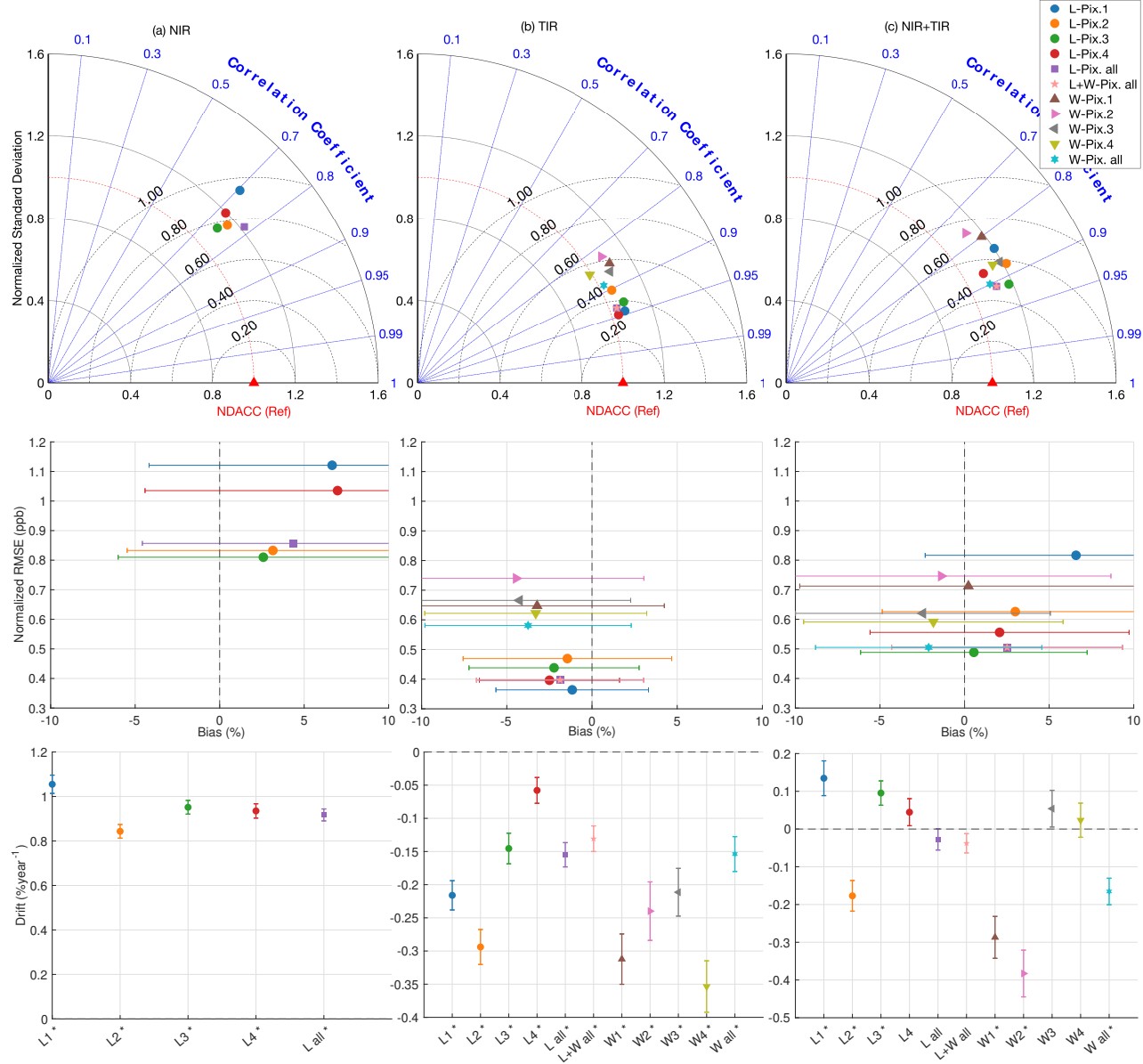

**Figure 11.** The normalized Taylor diagram (top row), and normalized RMSE versus average percent bias (middle row), and the drift (bottom row) for all MOPITT pixel measurements compared to NDACC. In the modified Taylor diagram, the normalized standard deviation (NSTD) is on the radial axis, the correlation coefficient value is on the angular coordinate, and the black dashed lines show the normalized CRMSE with respect to NDACC as the reference point. Column (a) shows the results for the NIR product, column (b) for the TIR product, and column (c) for the TIR-NIR. In row 2, the horizontal bars represent the 1-sigma standard deviation of the biases, and in row 3, the vertical bars are drift fit uncertainties (1-sigma). In row 3, the asterisks (∗) on the x-axis labels indicate drifts with significance levels of 95% or greater. The MATLAB SkillMetrics toolbox (https://github.com/PeterRochford/SkillMetricsToolbox, last retrieved September 1, 2019.) was used to create the Taylor diagrams.

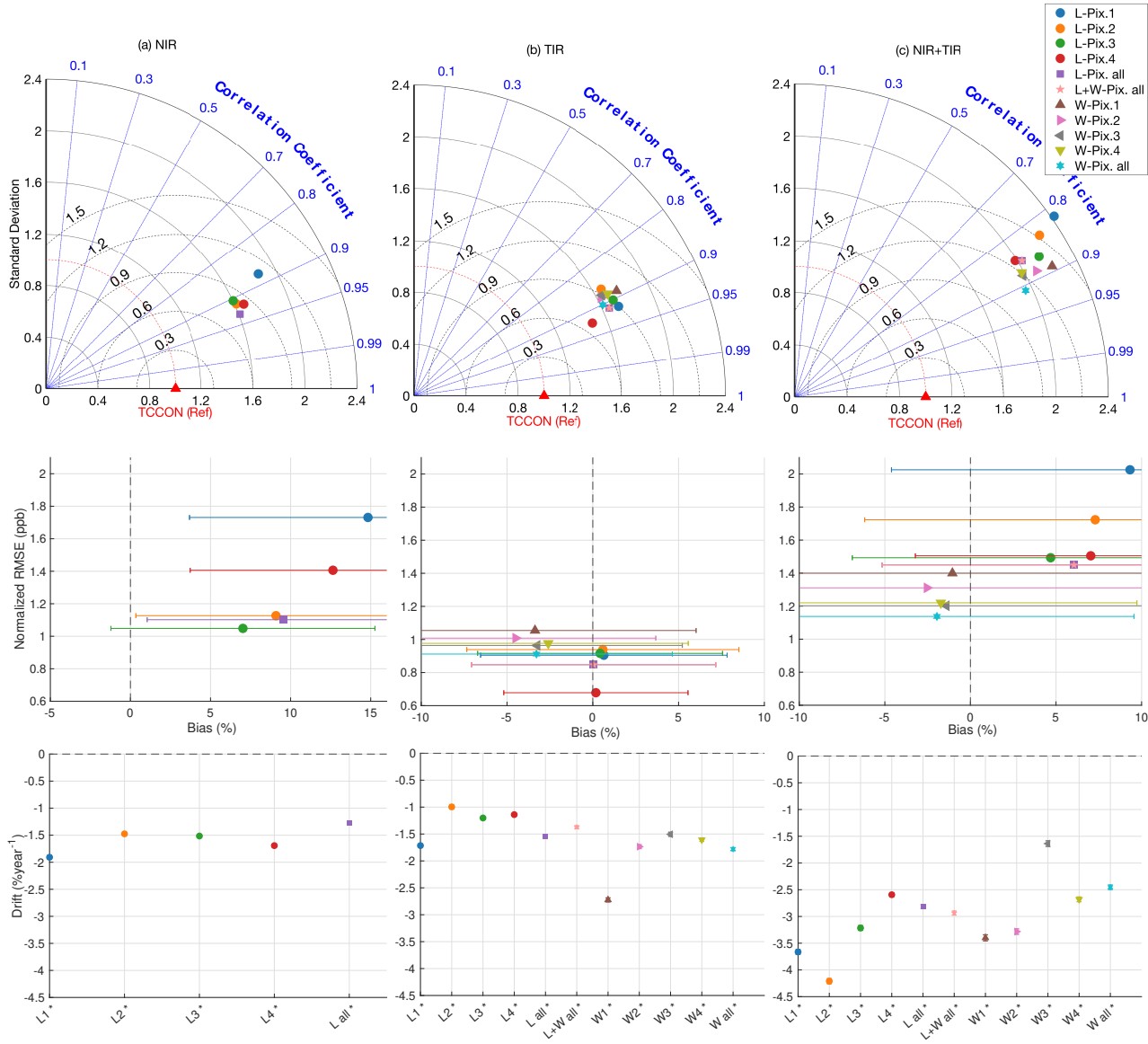

**Figure 12.** Same as Fig. 11 but for comparisons with TCCON measurements.

## 6.3 Comparison between NDACC and TCCON

The results in Fig. 11 and Fig. 12 show that the correlation coefficients between the MOPITT and TCCON measurements
$(0.8 - 0.95)$ are larger than those found between the MOPITT and NDACC measurements $(0.7 - 0.8)$ for the NIR product.





Similar results with respect to NDACC and TCCON are seen for the TIR product $(0.8 - 0.95)$ and for the joint TIR-NIR product $(0.8 - 0.9)$.

The NSTD in the comparison with NDACC measurements is generally between 1.0 and 1.2 for all pixels and all products. However, the NSTD values in the comparisons with TCCON measurements are increasing for each product from NIR (around

1.6) to TIR (between 1.6-1.8) and to joint TIR-NIR (between 1.8-2.5).

The pixels' bias and drift results reveal more information in the comparison between Fig. 11 and Fig. 12. The NIR product pixel biases for both NDACC and TCCON are positive and values vary from approx 3 to 7% for NDACC and approx 7 to 15% for TCCON. Pixel 1 is larger in TCCON by 5% and has the largest bias as well as normalized RMSE. For the TIR product, both figures show that all biases over water are negative and approximately -3%. The corresponding biases versus NDACC over land

are also negative and around -2%, however, the biases over land when compared with TCCON are around 1%. The magnitude of the bias values for the joint TIR-NIR product are similar in both the NDACC and TCCON comparisons. However, all the biases over water are negative for both NDACC and TCCON and they are positive over land. The joint TIR-NIR pixel biases over water are around $-2\%$ for both NDACC and TCCON, with the exception of the NDACC comparison for pixel 1, which is around 0%. The pixel biases over land are between 1% and 6% for the NDACC and between 4% and 15% for TCCON

datasets. Overall, pixel 1 over land has the largest bias.

The drift results for the NIR product for NDACC and TCCON are with opposite sign. The minimum drift of all the products between NDACC and TCCON is found for the TIR and joint TIR-NIR NDACC comparisons. For the TIR product, the drift values versus both NDACC and TCCON are negative, with those found with NDACC being much smaller than those with TCCON. In the TIR, the drifts for pixel 1 over water for the TCCON are the largest drifts among all the pixels. For the joint

TIR-NIR product, the drifts for all pixels are negative for TCCON, but are generally positive with NDACC for the pixels over land (except pixel 2 and all pixels) and are negative for pixels over water (except pixels 3 and 4). In the joint TIR-NIR product, all the drift values with NDACC are smaller than those with TCCON. The drift uncertainties for the NDACC comparisons are larger than those for TCCON because the number of comparison points used for NDACC and MOPITT is much smaller than those used for TCCON and MOPITT.

**6.4   Comparison with previous studies**

Previous MOPITT validation studies that included Arctic comparisons are Buchholz2017, Hedelius2019, and Deeter et al. (2019). In this section, their results are compared with this study. Buchholz2017 compared MOPITT V6 data from all three retrieval products with 14 ground-based NDACC FTIR sites from around the globe. They found that overall, pixel 1 has the largest bias and the smallest correlation coefficient among the pixels, and pixel 3 has the largest R value for all products over

land. In our study, pixel 1 has the lowest correlation coefficient in the NIR and joint TIR-NIR products (see Fig.11) as well as high bias values. However we find that pixel 4 has correlation and bias values in the NIR that are comparable to those of pixel 1. Unlike the NIR and joint TIR-NIR products, pixel 1 has a large correlation coefficient with low bias in the TIR. We also find that pixel 3 has also the largest correlation coefficient and smallest bias only in the joint TIR-NIR product. We should consider that these overall results presented in Buchholz2017 are the average for all sites. The results for the Eureka FTIR and MOPITT





V6 comparisons for each product are presented in Tables 3 and 4 of Buchholz2017. A summary of their results using MOPITT V6 compared to our study using V8 is given in Table 2. The correlation coefficients for V8 are higher than those for V6 for all products with the exception for the NIR over land. The reason could be because the number of measurements that we used in our V8 comparisons is almost three times that used by Buchholz2017 for V6. The bias standard deviations are larger for the V6

study than the V8 study, again except for the NIR product over land. However, all the biases for the V6 and V8 studies agree within their uncertainties. The bias values for the NIR and joint TIR-NIR products over land are equal, however, the MOPITT V8 TIR biases are larger than the V6 for both land and water. For the joint TIR-NIR product over water, the magnitudes of the V6 and V8 biases are the same but the signs are opposite. The drift uncertainties for all V6 results are much larger than those for V8 and the drift values of all V8 results are smaller than V6 except for the NIR over land. This is because of the known

issue in the MOPITT V6 products that was explained in Sect. 2. In V6, $N_2O$ was considered constant with time, which leads to larger drift values with time. This issue is solved in V7 and V8.

**Table 2.** Comparison between results presented in Buchholz2017 for MOPITT V6 data for the Eureka NDACC FTIR between 2006 and 2014 with results presented in this study for MOPITT V8 with Eureka NDACC FTIR data between 2006 and 2019.

| Field | Product | Version | Bias (%) | Drift ($\%\text{year}^{-1}$) | Correlation coefficient (R) | # observations |
|-------|---------|---------|----------|-------|-------------------|-------|
| Land | NIR | V6 | $4.36 \pm 7.37$ | $0.65 \pm 0.17$ | 0.92 | 889 |
| | | V8 | $4.35 \pm 8.9$ | $0.91 \pm 0.03$ | 0.79 | 2966 |
| | TIR | V6 | $0.7 \pm 8.66$ | $0.3 \pm 0.11$ | 0.72 | 1080 |
| | | V8 | $-1.9 \pm 4.9$ | $-0.15 \pm 0.02$ | 0.93 | 2602 |
| | NIR+TIR | V6 | $3.06 \pm 9.47$ | $0.2 \pm 0.23$ | 0.88 | 880 |
| | | V8 | $2.5 \pm 6.8$ | $-0.03 \pm 0.03$ | 0.91 | 2589 |
| Water | TIR | V6 | $0.3 \pm 8.22$ | $-0.87 \pm 0.13$ | 0.78 | 700 |
| | | V8 | $-3.8 \pm 6.0$ | $-0.15 \pm 0.03$ | 0.89 | 1952 |
| | NIR+TIR | V6 | $2.82 \pm 12.77$ | $-1.07 \pm 0.35$ | 0.62 | 580 |
| | | V8 | $-2.1 \pm 6.7$ | $-0.2 \pm 0.03$ | 0.90 | 2119 |

Hedelius2019 compared the MOPITT V7 joint TIR-NIR products with TCCON measurements globally between 2002 and 2018. As part of their dataset, they used Eureka TCCON measurements for the period between 2010 and 2018. Overall, they found that MOPITT TIR-NIR measurements are higher than TCCON by an average of 6.4%, or 3-10 ppb. They reported a

$6\pm5\%$ bias versus the Eureka FTIR measurements over land and a $8\pm10\%$ bias versus the Ny-Ålesund FTIR at 78° N over water. Our analysis shows that the MOPITT V8 joint TIR-NIR products for different pixels over land are between 1 and 7% (1 to 6 ppb) larger than the TCCON measurements. However, our results show a negative bias between $-1$ and $-4\%$ for MOPITT





V8 over water in comparison with Eureka TCCON measurements. Hedelius2019 only compared with MOPITT measurements over land near Eureka.

Deeter et al. (2019) compared the MOPITT V7 and V8 data for each product with NOAA (National Oceanic and Atmospheric Administration) in situ flask data sampled by aircraft. The NOAA measurements are made over North America and most of the locations in the USA. Their comparison period was from March 2000 until the end of 2018 with 1339 profiles in total used for all sites. The summary of their comparison is presented in Table 3 of Deeter et al. (2019). As bias values and their standard deviations for the total columns are reported in units of $molec.cm^{-2}$, it was necessary to convert their results to percent in order to compare with our results. For this purpose we consider an average total air column density of $2.0 \times 10^{25}$ $molec.cm^{-2}$ and 120 ppb for the annual average CO concentration over the northern hemisphere (EPA, 2000). The results are shown in Table 3. The two most northern NOAA measurement locations are the Poker Flat station in Alaska (65.07 °N) and the East Trout Lake station in Saskatchewan, Canada (54.35 °N). The majority of the data collected for these two sites are between 2006 and 2012 with a few data points out of this range. The results for these two northern stations are presented separately in Table 3. Almost all of the NOAA measurements are over land, therefore for this comparison we used only MOPITT pixels over land.

As shown in Table 3, the bias values for each of the MOPITT products compared with all NOAA measurement sites are smaller for V8 than V7 (however with opposite sign for the NIR). For the NIR and joint TIR-NIR V8 products, the biases increase poleward when considering all NOAA sites, the two northern sites and our Eureka results. The TIR bias has a different pattern than the other two products. For the MOPITT V8 TIR product, the bias relative to all NOAA sites is equal to that for the two northern NOAA sites. However, for Eureka, the TIR bias for comparisons with NDACC is negative and its magnitude is larger than that versus all NOAA sites. In contrast, the bias from the TIR - TCCON comparisons is smaller than that for all NOAA sites and has the same sign. However, the standard deviations of all these biases are large and they agree within their combined uncertainties. Table 3 also shows that all uncertainties for the V7 comparisons with all NOAA sites are larger than for V8 and their magnitudes are almost twice as large. The comparison between the calculated drift values is also reported in Table 3. The drifts versus all NOAA sites for V7 and V8 have opposite signs with a much larger magnitude drift in V7 for all products. Comparing the MOPITT V8 drift values from the different stations reveals that almost all of the TIR and joint TIR-NIR product drifts are negative and the NIR has a positive drift value. For the NIR product, the drift values increase with increasing latitude for all V8 comparisons. The smaller biases and drifts versus all NOAA sites found for MOPITT V8 compared to V7 are due to the improvement in radiance bias correction parameterization applied in V8. Comparing the uncertainties for the drift values shows the same pattern as for the bias values, with the uncertainties for MOPITT V7 being larger than those for V8. The drift uncertainties for the comparison with the Eureka FTIR measurements are much smaller than those versus the NOAA northern sites, which could be because of the larger number of measurements used in our comparison. The correlation coefficients for all NOAA sites with MOPITT V8 are larger than those for V7. These values decrease slightly (by 0.03 to 0.07) when considering only the northern NOAA sites. The correlation coefficients found for MOPITT V8 for the Eureka comparisons are higher than those for the NOAA sites, which again could be because of the larger number of measurements that were used in this study than in Deeter et al. (2019). Overall, a significant improvement in MOPITT V8 biases, drift values and correlations can be



**Table 3.** Comparison between NIR, TIR and joint TIR-NIR results presented in Table 3 of Deeter et al. (2019) for MOPITT V7 and V8 data for all NOAA stations (rows 1 and 2) and for the two most northern NOAA (row 3) stations between 2000 and 2018, along with results from this study for MOPITT V8 (all pixels over land) with NDACC (row 4) and TCCON (row 5). The unit of bias is % and for drift is %year$^{-1}$. The northern NOAA measurements are for the Poker Flat station in Alaska (65.07 °N) and the East Trout Lake station in Saskatchewan (54.35 °N). The northern results are based on Deeter et al. (2019) (M. N. Deeter, personal communication). The uncertainties presented in rows 4 and 5 for bias values compared with NDACC and TCCON are the standard deviation of the biases (1 sigma).

| Product | Measurements | Bias (%) | Drift (%year$^{-1}$) | Correlation coefficient (R) |
|---------|--------------|----------|----------------------|------------------------------|
| NIR | V7 NOAA all sites | $-0.4 \pm 10$ | $-1.01 \pm 0.26$ | 0.04 |
|  | V8 NOAA all sites | $0.4 \pm 5.4$ | $0.05 \pm 0.11$ | 0.60 |
|  | V8 NOAA northern sites | $2.0 \pm 5.4$ | — | 0.57 |
|  | V8 NDACC Eureka | $4.3 \pm 8.9$ | $0.91 \pm 0.03$ | 0.79 |
|  | V8 TCCON Eureka | $9.0 \pm 8.0$ | $-1.30 \pm 0.01$ | 0.94 |
| TIR | V7 NOAA all sites | $3.0 \pm 11.5$ | $0.77 \pm 0.34$ | 0.58 |
|  | V8 NOAA all sites | $0.83 \pm 5.8$ | $-0.02 \pm 0.05$ | 0.82 |
|  | V8 NOAA northern sites | $0.83 \pm 5.0$ | — | 0.79 |
|  | V8 NDACC Eureka | $-1.8 \pm 4.9$ | $-0.15 \pm 0.02$ | 0.93 |
|  | V8 TCCON Eureka | $0.19 \pm 8.1$ | $-1.52 \pm 0.02$ | 0.91 |
| NIR+TIR | V7 NOAA all sites | $2.5 \pm 10.8$ | $-1.08 \pm 1.80$ | 0.57 |
|  | V8 NOAA all sites | $0.8 \pm 6.6$ | $0.001 \pm 0.070$ | 0.81 |
|  | V8 NOAA northern sites | $1.6 \pm 2.5$ | — | 0.74 |
|  | V8 NDACC Eureka | $2.5 \pm 6.8$ | $-0.03 \pm 0.03$ | 0.91 |
|  | V8 TCCON Eureka | $6.0 \pm 11.1$ | $-2.80 \pm 0.03$ | 0.87 |

seen in comparison with V7 in Table 3. The results from this study show that these biases generally increase with increasing latitude.

# 7 Summary and conclusions

Previously, several global studies have validated MOPITT CO data using ground-based FTIR measurements from either NDACC or TCCON. Their results indicated that there is a large bias in MOPITT V6 and V7 data above 60°N, which makes using these data difficult in that region. The latest version of MOPITT retrieval products is V8. This study has validated the MOPITT V8 CO total column measurements by comparing to both NDACC and TCCON CO total column measurements in the Canadian high Arctic.





This study and others have investigated the MOPITT pixel biases. Deeter et al. (2015), using MOPITT V6 data, found that pixel 3 has the largest instrumental noise. Buchholz2017 showed that pixel 1 has the largest positive bias globally in the MOPITT V6 data. Finally, Hedelius2019 showed that pixel 1 has the largest negative bias and that biases increase poleward in MOPITT V7. Our results for MOPITT V8 show that pixel 1 has the largest bias among all 4 pixels over the Arctic, which

agrees with Hedelius2019. Our monthly pixel bias investigation (Fig. 1) reveals that there is a bias in the summer months in all pixels. Figure 2 illustrates that the bias in those months is likely due to the mixture of ice and water over the ocean and patchy snow over the land. Another result of the monthly pixel bias investigation is that pixel 1 measurements over land have a large systematic bias that could induce bias into multi-pixel averages for V8. The pixel 1 bias over water is similar to the biases of the other pixels. We can conclude that there is a systematic bias in pixel 1 over land. Pixel 3 also has a systematic positive bias

over land in the spring months.

We compared the CO profile and total column averaging kernels for the MOPITT and FTIR retrievals as they have different vertical resolutions. We also analyzed the DOFS for the three MOPITT products and the NDACC measurements. The MOPITT column AKs over water are greater in the mid-troposphere than those over land, especially for the TIR products. This is because the thermal contrast is smaller over water and there is no contribution from the lower troposphere. TCCON and NDACC AKs

showed more sensitivity to changes in the upper troposphere and above, however, MOPITT retrievals are typically more sensitive to the mid-troposphere. The MOPITT TIR product is more sensitive to the mid-troposphere and the joint TIR-NIR product is more sensitive to the mid- and lower troposphere.

After accounting for the difference of averaging kernels for the instruments, we compared the MOPITT CO measurements to the NDACC and TCCON retrievals by separating the MOPITT results by pixel, land type, and data product. Before running

the comparisons, we applied a filter to the MOPITT data to reduce the effects of outliers. In order to simplify and visualize the comparisons between different combinations of pixels and land types, we used modified Taylor diagrams as well as plots of bias and drift to evaluate the biases, uncertainties, and correlation coefficients between MOPITT V8 and the Eureka FTIR measurements.

Our results show that there is good consistency between the MOPITT-NDACC and MOPITT-TCCON CO comparisons. The

comparisons of the MOPITT V8 measurements with NDACC and TCCON show that the bias values are generally positive for the NIR and negative for the TIR and joint TIR-NIR products. However, the biases and drifts versus NDACC for the TIR and joint TIR-NIR products are smaller than those versus TCCON. Pixel 1 has the largest bias in the NIR and joint TIR-NIR products in both the TCCON and NDACC comparisons; however, pixel 1 shows good performance in the TIR product comparisons for both NDACC and TCCON. We recommend to use only TIR measurements from pixel 1 in the high Arctic. For

the TIR product, the bias and drift values are larger over water than over land when compared to both NDACC and TCCON. However, the bias values are generally less than 5%. The TIR drift values versus TCCON are twice as large as those versus NDACC. In the joint TIR-NIR products, all pixels' biases over land are positive and are negative over water for both the NDACC and TCCON comparisons.

Finally, we compared our results with other studies for the three latest versions of MOPITT data both globally and regionally.

There is a good consistency between our total column bias comparison for MOPITT V8 vs. NDACC with the MOPITT V6





biases from Buchholz2017 (Table 2) for the NIR and joint TIR-NIR products. However, this consistency is not seen for the TIR products. There is low thermal contrast in the Arctic region and the DOFS are generally low (Fig. 7). The average TIR DOFS is less than 1 for the Eureka station. Therefore, the contribution of *a priori* information is high in the retrievals. The difference in our biases in the TIR product comparisons is due to the improvements applied to the MOPITT V8 retrieval relative to V6.

Our MOPITT vs. TCCON comparison is generally consistent with Hedelius2019. The MOPITT joint TIR-NIR products are greater than TCCON by around 6-8 % globally based on Hedelius2019, and based on this study are $4-9\%$ (depending on pixel) for the Arctic. A similar comparison with the total column results of Deeter et al. (2019) revealed that there is a correlation between the total column biases with latitude; larger biases were observed at higher latitudes. We also observed that consistent bias results were found between this study and that of Deeter et al. (2019) (Table 3) for the two northern sites around 60 °N.

Generally, the DOFS of MOPITT measurements in the Arctic is small (average around 1) because of the low thermal contrast. Compared to MOPITT V6 and V7, our comparisons in the Canadian high Arctic show that there are significant improvements in MOPITT V8. In addition to the enhancements in the V8 retrievals, using a filter to reduce the effect of outliers in the Arctic region improved our comparisons with ground-based FTIR measurements from Eureka, Nunavut.

*Data availability.* MOPITT Version 8 data are freely available at https://earthdata.nasa.gov/. The NDACC PEARL FTIR CO measure-
ments are available from the NDACC database hosted by NOAA at https://www-air.larc.nasa.gov/missions/ndacc/data.html and the TCCON PEARL FTIR measurements used in this work are available at https://doi.org/10.5683/SP3/1GBGMY.

*Author contributions.* AJ gathered the datasets and conducted comparisons between the datasets, created the plots and tables, and wrote the manuscript. KAW advised and supervised the work and provided comments, advice, and editing. RB and DW provided guidance regarding previous validation results and methodologies. MD provided advice and the results in Table 3 for the northern sites. KS, EL, and TW
provided the PEARL NDACC retrievals. KS and SR provided the PEARL TCCON data. PF contributed to collecting TCCON and NDACC measurements at PEARL. JRM supported the operations of PEARL and MOPITT. HMW supported the MOPITT data and operation. All coauthors provided comments and contributed to editing the manuscript. EM provided assistance with the TCCON data preparation.

*Competing interests.* One co-author is a member of the editorial board of this journal. The authors declare that they have no other competing interests to declare.

*Acknowledgements.* The Bruker FTIR measurements were made at PEARL by CANDAC. CANDAC has been supported by the Atlantic
Innovation Fund/Nova Scotia Research Innovation Trust, Canada Foundation for Innovation, Canadian Foundation for Climate and Atmospheric Sciences (CFCAS), Canadian Space Agency (CSA), Environment and Climate Change Canada (ECCC), Government of Canada International Polar Year funding, Natural Sciences and Engineering Research Council (NSERC), Northern Scientific Training Program



(NSTP), Ontario Innovation Trust, Polar Continental Shelf Program, and Ontario Research Fund. The MOPITT team acknowledges support from the CSA, NSERC, and ECCC, and the contributions of COMDEV (the prome contractor and ABB BOMEM. The NCAR MOPITT project is supported by the National Aeronautics and Space Administration (NASA) Earth Observing System (EOS) Program. The National Center for Atmospheric Research (NCAR) is sponsored by the National Science Foundation. The authors would like to thank Jacob K.
5   Hedelius for his guidance on previous validation results. AJ and KAW would like to thank Shannon Hicks-Jalali for assistance editing the paper.





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
