# Peer review of "A comparison of carbon monoxide retrievals between the MOPITT satellite and Canadian High-Arctic ground-based NDACC and TCCON FTIR measurements"

_Atmospheric Measurement Techniques, 2022_

## Referee Comment (RC1)

Review comments on the preprint: „A comparison of carbon monoxide retrievals between the MOPITT satellite and Canadian High-Arctic ground-based NDACC and TCCON FTIR measurements" by Ali Jalali et al.

**General comments:**

The manuscript compares MOPITT CO v8 retrievals from 2006-2019 with TCCON and NDACC FTIR observations from the PEARL Eureka site. MOPITT observations within 110km and for 24h intervals are compared. The authors thoroughly describe the methodologies for comparing space-based total column retrievals with those ground-based networks. Beside the pixel-to-pixel bias the work examines and compares MOPITT NIR, TIR and NIR-TIR retrievals with the ground-based site and provides comprehensive of statistics. The results and improvements of v8 are thoroughly discussed and described.

- consider to mention that the fundamental and first overtone of CO are relevant in the TIR/NIR spectral region

- do I get this right: each channel has four pixles, and each pixel in a channel sounds a certain part of a single CO line? if so does that mean that the inner pixel record transmittance from the line and the outer pixel from the line wing? why not sensing mutliple CO lines?

**Specific comments:**

Page3, line 2:
Consider to include the MAPS mission aboard the space shuttle (e.g. Reichle 1999)

p3,18:
Does NDACC only measure absorption of solar radiation or does the network also measure emission from the atmosphere?

p4,21: Maybe add one sentence why log(vmr) is fitted?

p4,23: One or two words explaining the ‚outer' and ‚inner' pixels meaning (line centre vs line wing?). Also see general comment above.

p5,33 and Fig.1:
Just for confirmation, so the weighted average of all pixels means that each pixel's average within the 30day/100km was multiplied by 0.25 and then summed?

p5, Fig1:
Is a joint TIR-NIR retrieval possible over water? does the signal then basically only come from the TIR channel?

p9,4:
Maybe add one sentence on how well the assumption for the small-area approximation is fulfilled at the Eureka site (to justify the chosen 1° radius criteria).

p9,16 and Fig. 4:
It is mentioned (in p13,30) that DOFS represent the information content of the retrieval,
so why do high DOFS lead to large delta xCO?
Does it mean that the priors are already close to the true values?
Are low DOFS retrievals trustworthy fits?
Please clarify the meaning of DOFS for the present work.

p10, Fig. 4:
Why are the SZA values ranging from 60-120°? I would expect it from 60-90°.
Also check p16,5 which designates daytime measurements (SZA < 90°).
Note that the figure caption says the RMS is represented in blue when it actually is yellow.

p12,20:
Is there a reason why NDACC is not using 4 times daily (6 hourly) NCEP data?

p13, Sec.4:
Consider to include a sentence which states that the total column averaging kernel is
computed by a total column operator (converts profile concentrations to a column
concentration, see Deeter 2002)

p18,3:
what is the motivation for 500 grid points per MOPITT layer? just to be dense enough?

p18,13:
What is the motivation for selecting a critical difference of 80hPa?
is it proposed somewhere (in Kerzenmacher et al. 2012?)

p18,27:
Consider to explain why MOPITT retrievals are in log space (see previous comment p4,21)

p19,,11:
If TCCON method IV from Hedelius is similar to the method used for NDACC (more
consistent?), why not using it instead of method II?

p20, Eq.(10):
Why is the number of measurements N the same for the MOPITT and FTIR (because of the
averaged MOPITT measurements?)

p22,5:
is the larger correlation to NDACC caused by the TIR interval?
is the Fig. 9 plot also available for MOPITT NIR?

p29, Sec. 6.4:
likely different co-location and filter criteria across studies are responsible for some of the difference (as you mention three times more V8 comparisons than V6 in Buchholz).
Consider mentioning the effect of coincide criteria w.r.t. Hedelius 2019 in paragraph p29,12.

**Technical corrections:**

p31, Table 3:
Is the entry for the `V7 NOAA all sites` correlation coefficient a typo?

Write text in equations upright in order to discriminate it from variables.

---

## Author Comment (AC1)

**Response to Referee #1 – "A comparison of carbon monoxide retrievals between the MOPITT satellite and Canadian High-Arctic ground-based NDACC and TCCON FTIR measurements" by Ali Jalali et al.**

We would like to thank Referee #1 for their helpful comments. Here we address their review, with their comments in green and our responses indented in black.

**General comments:**

The manuscript compares MOPITT CO v8 retrievals from 2006-2019 with TCCON and NDACC FTIR observations from the PEARL Eureka site. MOPITT observations within 110km and for 24h intervals are compared. The authors thoroughly describe the methodologies for comparing space-based total column retrievals with those ground-based networks. Beside the pixel-to-pixel bias the work examines and compares MOPITT NIR, TIR and NIR-TIR retrievals with the ground-based site and provides comprehensive of statistics. The results and improvements of v8 are thoroughly discussed and described.

- consider to mention that the fundamental and first overtone of CO are relevant in the TIR/NIR spectral region

We have added the following comment to Sec. 3 (FTIR Instrument) on p12,10. "The FTIR measures CO absorption spectra from either the fundamental band (NDACC) or first overtone band (TCCON)."

- do I get this right: each channel has four pixles, and each pixel in a channel sounds a certain part of a single CO line? if so does that mean that the inner pixel record transmittance from the line and the outer pixel from the line wing? why not sensing mutliple CO lines?

The four MOPITT pixels provide spatial information not spectral information. They are arranged linearly in the orbit "along-track" direction. As the instrument is a gas correlation radiometer, the spectrum is not measured directly.

**Specific comments:**

Page3, line 2:
Consider to include the MAPS mission aboard the space shuttle (e.g. Reichle 1999)

We have made this update to p3, l2:

From: "Over the past two and a half decades, CO has been measured from space using a suite of nadir sounders. One of the earliest satellite-based instruments that measured CO was the Interferometric Monitor for Greenhouse Gases (IMG) (1996) (Wang et al., 1998) which collected eight months of data in 1996-1997."

To: "Over the past four decades, CO has been measured from space using a suite of nadir sounders. The earliest instruments that measured CO were Measurements of Air Pollution from Satellites (MAPS) which flew on the Space Shuttle in 1981, 1984 and 1994 (Reichle et al., 1999) and the Interferometric Monitor for Greenhouse Gases (IMG) (Wang et al., 1998) which collected eight months of data in 1996-1997."

Reference:

Reichle, H. G., et al.: Space shuttle based global CO measurements during April and October 1994, MAPS instrument, data reduction, and data validation, J. Geophys. Res., 104, 21 443–21 454, doi:10.1029/97JD03299, https://doi.org/10.1029/97JD03299, 1999.

p3,18:
Does NDACC only measure absorption of solar radiation or does the network also measure emission from the atmosphere?

The NDACC FTIR instruments only measure solar absorption spectra.

p4,21: Maybe add one sentence why log(vmr) is fitted?

This following sentence has been added on p4,22. "Compared to retrievals of VMR, the log(VMR)-based retrieval algorithm improves retrieval convergence and yields fewer profiles with unphysically small VMR values (Deeter et al., 2007)."

Reference:
Deeter, M. N., Edwards, D. P., and Gille, J. C. (2007), Retrievals of carbon monoxide profiles from MOPITT observations using lognormal a priori statistics, *J. Geophys. Res.*, 112, D11311, doi:10.1029/2006JD007999.

p4,23: One or two words explaining the ‚outer' and ‚inner' pixels meaning (line centre vs line wing?). Also see general comment above.

As mentioned above, the MOPITT pixels provide spatial information.  This has been clarified in the text as follows.

From:  "Each channel's detector is comprised of a four-pixel linear array, where 1 and 4 are the outer pixels and 2 and 3 are the inner pixels of the array."

To:  "Each channel's detector is comprised of a four-pixel linear array oriented along-track, where 1 and 4 are the outer pixels and 2 and 3 are the inner pixels of the array."

p5,33 and Fig.1:
Just for confirmation, so the weighted average of all pixels means that each pixel's average within the 30day/100km was multiplied by 0.25 and then summed?

No, all CO measurements from all pixels with their corresponding uncertainties (within 110 km radius and 30 days) are used to calculate the weighted average. Equations 4.20 to 4.23 from Bevington 1969 are used to calculate the weighted average.

Bevington, P. R., Data Reduction and Error Analysis for the Physical Sciences, 336 pp., McGraw-Hill, 1969.

p5, Fig1:
Is a joint TIR-NIR retrieval possible over water? does the signal then basically only come from the TIR channel?

Yes, as mentioned on p4,30 MOPITT NIR retrieval is only made over land. Therefore, for pixels over water, the joint TIR-NIR retrieval information is coming from TIR only.

p9,4:
Maybe add one sentence on how well the assumption for the small-area approximation is fulfilled at the Eureka site (to justify the chosen 1° radius criteria).

The 1° is consistent with the ~100 km x 100 km region assumed as sufficiently small by Hedelius2019 in their work.  This has been clarified as follows on p9,4:

From:  "…based on the assumption that over a small enough area (1° radius)…"

To:  "…based on the assumption that over a small enough area (~100 km x 100 km or 1° radius, used here)…"

p9,16 and Fig. 4:
It is mentioned (in p13,30) that DOFS represent the information content of the retrieval, so why do high DOFS lead to large delta xCO?
Does it mean that the priors are already close to the true values?

There are a few factors that could be influencing to the larger $\Delta X_{CO}$ for higher DOFS. First, there are a lower number of retrievals with higher DOFS and these make up are less than 5 percent of total retrievals. Second, the larger $\Delta X_{CO}$ is due mainly to pixel 1 (which has the largest systematic bias).

It may be counterintuitive, but it is not unexpected that increasing DOFS will be associated with larger retrieval biases. As DOFS decreases, retrieved profiles are more heavily weighted by the a priori. So, as DOFS decreases, retrievals for all four pixels will tend toward the a priori, and any inter-pixel biases will go to 0. DOFS is a useful index for the information content (or weighting of the measured radiances in the retrieval), not 'retrieval quality'. Information content and retrieval bias are really separate concepts.

Are low DOFS retrievals trustworthy fits?
Please clarify the meaning of DOFS for the present work.

The MOPITT v8 user guide recommended not to filter any data due to characteristics of the retrieval averaging kernels like DOFS. As mentioned in the paper, we did not use DOFS as a filter criterion.

DOFS is a retrieval parameter output provided for each of the datasets and is defined on p13,30-31.

p10, Fig. 4:
Why are the SZA values ranging from 60-120°? I would expect it from 60-90°.

The MOPITT TIR measurements do not require sunlight so the instrument provides measurements in these channels up to SZA of 120° (for the data set we examined within 110 km of PEARL).

Also check p16,5 which designates daytime measurements (SZA < 90°).

This is correct. To be consistent with the previous study by Buchholz2017, we only used daytime measurements when identifying coincident measurements for the comparison.

Note that the figure caption says the RMS is represented in blue when it actually is yellow.

This has been fixed in the paper. The RMS symbols are green (olive).

p12,20:
Is there a reason why NDACC is not using 4 times daily (6 hourly) NCEP data?

NDACC uses the daily profiles at 1200 GMT that NCEP generates for all NDACC sites as described in the metadata file located here: https://www-air.larc.nasa.gov/pub/NDACC/PUBLIC/meta/ncep/ncep_2022.pdf.

This has been clarified in the text as follows:

 "The National Centers for Environmental Protection (NCEP) provides daily temperature and pressure profiles at 1200 GMT interpolated to the geographical location of NDACC stations; those for Eureka are used in the retrieval (https://www-air.larc.nasa.gov/missions/ndacc/data.html?NCEP=ncep-list)."

In addition, we have changed the original sentence on p12,19.

From: "The mean outputs from Whole Atmosphere Chemistry Climate Model (WACCM) version 4 between 1980-2020 are used for the a priori VMR profiles (Marsh et al., 2013) and daily temperature and pressure profiles from the National Centers for Environmental Protection (NCEP) interpolated to

the geographical location of PEARL are used in the retrieval
(ftp://ftp.cpc.ncep.noaa.gov/ndacc/ncep/).”

To: “The mean outputs from Whole Atmosphere Chemistry Climate Model (WACCM) version 4 between 1980-2020 are used for the a priori VMR profiles (Marsh et al., 2013). The National Centers for Environmental Protection (NCEP) provides daily temperature and pressure profiles at 1200 GMT interpolated to the geographical location of NDACC stations; those for Eureka are used in the retrieval (https://www-air.larc.nasa.gov/missions/ndacc/data.html?NCEP=ncep-list).”

p13, Sec.4:
Consider to include a sentence which states that the total column averaging kernel is computed by a total column operator (converts profile concentrations to a column concentration, see Deeter 2002).

This sentence has been added to the paper on p13,14. “The CO total column averaging kernel is calculated from the profile averaging kernel matrix and total column operator (Deeter, 2002).”

Deeter, M. (2002): Calculation and Application of MOPITT Averaging Kernels, https://www.acom.ucar.edu/mopitt/avg_krnls_app.pdf, last accessed 17 Sept. 2022.

p18,3:
what is the motivation for 500 grid points per MOPITT layer? just to be dense enough?

Yes, this was increased to 500 points to ensure that there were sufficient grid points per MOPITT layer for the interpolation.

p18,13:
What is the motivation for selecting a critical difference of 80hPa? is it proposed somewhere (in Kerzenmacher et al. 2012?)

When Buchholz2017 applied the method from Kerzenmacher et al. (2012), they used a critical distance of 20 hPa for most locations but increased this value to 50 hPa for high latitude and altitude stations such as PEARL. We have used a slightly higher value to increase the number of profiles included in the comparison dataset.

p18,27:
Consider to explain why MOPITT retrievals are in log space (see previous comment p4,21)

This information was added on p4,21. See above.

p19,,11:
If TCCON method IV from Hedelius is similar to the method used for NDACC (more consistent?), why not using it instead of method II?

We chose to use methods consistent and similar to previous studies for each network in our study.

p20, Eq.(10):
Why is the number of measurements N the same for the MOPITT and FTIR (because of the averaged MOPITT measurements?)

Yes, this is correct.

p22,5:
is the larger correlation to NDACC caused by the TIR interval? is the Fig. 9 plot also available for MOPITT NIR?

Figure 9 presents an example pair of correlation plots for the NDACC and TCCON comparisons. The plots for all pixel and channel combinations are given in the supplementary document. The range of correlation values for NDACC and TCCON with the MOPITT TIR channel can be seen in

panel (b) of Figures 11 and 12, respectively. The average for all pixels is fairly consistent between NDACC and TCCON with a greater spread in values for NDACC comparisons.

p29, Sec. 6.4:
likely different co-location and filter criteria across studies are responsible for some of the difference (as you mention three times more V8 comparisons than V6 in Buchholz). Consider mentioning the effect of coincide criteria w.r.t. Hedelius 2019 in paragraph p29,12.

This sentence has been added to p30,1. "This could also be influenced by the looser coincidence criteria used by Hedelius2019 for high latitude stations (4° x 8° versus 1° radius used here)."

**Technical corrections:**

p31, Table 3:
Is the entry for the `V7 NOAA all sites` correlation coefficient a typo?

No, in Table 3 of Deeter et al. 2019, under V7N the total column correlation is 0.04.

Write text in equations upright in order to discriminate it from variables.

This has been done.

---

## Author Comment (AC2)

**Response to Referee #3 – "A comparison of carbon monoxide retrievals between the MOPITT satellite and Canadian High-Arctic ground-based NDACC and TCCON FTIR measurements" by Ali Jalali et al.**

We would like to thank Referee #3 for their helpful comments. Here we address their review, with their comments in green and our responses indented in black.

**General Comments**

In this new study, Jalali et al. discuss the validation of MOPITT v8 satellite measurements of carbon monoxide with high latitude ground-based FTIR measurements in the Canadian Arctic. Following a concise introduction and description of the MOPITT and FTIR ground-based measurements, the vertical sensitivity of the measurements of the different instruments in terms of averaging kernels and degrees of freedom for signal are discussed. The methodology section clearly outlines the comparison approach and properly introduces the Taylor diagram as a tool to summarize the intercomparisons of the various MOPITT data subsets with the NDACC and TCCON measurements. For the actual comparisons with the reference data, the focus is given to multiple aspects, including MOPITT pixel-to-pixel bias, noise, and drifts. Results are compared to earlier validation studies using MOPITT v6 and v7 data, illustrating the benefits of the improved retrieval method of the v8 data. Major results are nicely summarized in the conclusions.

Overall, I got the impression that this is a carefully conducted study with sound results, applying state-of-the-art methodologies (e.g., consideration of the averaging kernels in the comparisons and the Taylor diagrams). The study properly considers the results of earlier work on MOPITT CO retrieval validation. The authors clearly spent time and effort to prepare and submit such a well-written, clear, and concise manuscript. Overall, I have only a short list of clarifications and suggestions and would like to recommend the paper for publication in AMT.

**Specific Comments**

p1, l9: Suggest replacing "within a 1° radius" with "within 110 km radius" (as stated later in the manuscript).

> This has been done.

p2, l12: At the end of the abstract, perhaps add a sentence about any broader implications and/or an outlook of the study?

> We have added this sentence to the end of the abstract. "Overall, this study aims to provide detailed validation for MOPITT v8 measurements in the Canadian high Arctic."

p2, l13-33: In the introduction, it could be elaborated and referenced that the MOPITT CO measurements are of interest for chemistry-transport and climate model validation as well as studies of tropospheric tracer transport, I think. Model validation and transport studies will benefit from proper error characterization and improved accuracy of the new MOPITT v8 data, as described here.

> At the end of p2, l33, the following sentence has been added. "To predict future warming in the Arctic and simulate air pollution impacts in this region, well-validated atmospheric chemistry models are required and these must be evaluated using high-latitude measurements (e.g., Monks et al., 2015; Whaley et al., 2022)."

> References:

> Monks, S. A., et al.: Multi-model study of chemical and physical controls on transport of anthropogenic and biomass burning pollution to the Arctic, Atmos. Chem. Phys., 15, 3575–3603, https://doi.org/10.5194/acp-15-3575-2015, 2015.

Whaley, C. H., et al. Model evaluation of short-lived climate forcers for the Arctic Monitoring and Assessment Programme: a multi-species, multi-model study, Atmos. Chem. Phys., 22, 5775–5828, https://doi.org/10.5194/acp-22-5775-2022, 2022.

p3, l16-27: Did the earlier studies provide any reasons for biases between the NDACC and TCCON measurements?

These earlier studies investigated these differences in detail and found that a number of factors can contribute to these biases. These include the air-mass-independent correction factor used by TCCON, use of different a priori VMR profiles, choices of spectroscopic line lists used, and the impact of smoothing errors.

p4, 24: What are the actual pressure levels of the MOPITT retrievals (or maybe, what is the vertical range with meaningful retrieval results)?

This has been added to the sentence on p4,24 as follows.

Changed from: "…profiles are provided on a 10-level fixed-pressure grid as the average VMR within each layer, …"

To: " … profiles are provided on an equal spaced 10-level fixed-pressure grid (surface, 900 hPa, 800 hPa, ..., 100 hPa) as the average VMR within each layer, …"

p33, l10-13: At the end of the conclusions, perhaps add 1-2 sentences on the broader implications of the study and future work (similar to the abstract).

We have added the following sentences to the end of the conclusions. "Together, these filtering and pixel usage recommendations and comparison results provide guidance for using MOPITT v8 measurements for studies in the Canadian high Arctic.  The improvements seen in this latest data version for MOPITT are encouraging for studies using this dataset in the high northern latitudes."

**Technical Corrections**

p2, l23: remove "140pp"?

This has been done.

p4, l6-7: Would like to suggest not using "Buchholz2017" and "Hedelius2019" and simply keep the references in their original formatting. It does not look like this makes the paper any shorter.

In writing the paper, we chose to use these short forms to make it easier to follow for the reader and because these references are used in the paper 18 and 16 times, respectively.

p4, l9: NASA'_s_

Changed.

p4, l12: 22 x 22 km_^2_

This has been changed to 22 km x 22 km.

p5, l18: _the_ water vapor (?)

The word "that" in this sentence has been changed to "the" as follows: "The new parameterization includes the date and geographical location of the MOPITT observation and the water vapour total column at the observation time."

p9, Table 1: use Copernicus table layout

This has been done.